# Natural genetic variation quantitatively regulates heart rate and dimension

Jakob Gierten [1,2,3,12], Bettina Welz[1,3,4,12], Tomas Fitzgerald [5,12], Thomas Thumberger [1], Rashi Agarwal [1,4], Oliver Hummel[6], Adrien Leger [5], Philipp Weber[7], Kiyoshi Naruse [8], David Hassel[3,7], Norbert Hübner [6,9,10,11], Ewan Birney [5] ✉ & Joachim Wittbrodt [1,3] ✉

The polygenic contribution to heart development and function along the health-disease continuum remains unresolved. To gain insight into the genetic basis of quantitative cardiac phenotypes, we utilize highly inbred Japanese rice fish models, *Oryzias latipes*, and *Oryzias sakaizumii*. Employing automated quantification of embryonic heart rates as core metric, we profiled phenotype variability across five inbred strains. We observed maximal phenotypic contrast between individuals of the HO5 and the HdrR strain. HO5 showed elevated heart rates associated with embryonic ventricular hypoplasia and impaired adult cardiac function. This contrast served as the basis for genome-wide mapping. In an F2 segregation population of 1192 HO5 x HdrR embryos, we mapped 59 loci (173 genes) associated with heart rate. Experimental validation of the top 12 candidate genes by gene editing revealed their causal and distinct impact on heart rate, development, ventricle size, and arrhythmia. Our study uncovers new diagnostic and therapeutic targets for developmental and electrophysiological cardiac diseases and provides a novel scalable approach to investigate the intricate genetic architecture of the vertebrate heart.

Cardiac phenotypes have specific morphological and functional hallmarks that can be assessed quantitatively and are typically not binary; instead, they manifest along a continuum that stretches from healthy to pathological forms. This causes a dilemma for both diagnosis and research into the early stages of the disease because normal variability between individuals examined pre- or postnatally can mask quantitative phenotypes. For example, in congenital heart disease (CHD), left ventricular (LV) hypoplasia is a quantitative phenotype ranging from mildly reduced ventricular size to a diminutive left ventricle observed in hypoplastic left heart syndrome (HLHS)[1]. While the pathophysiological description of HLHS dates back to 1851[2], its hereditary nature[3] has been a moving target as the genetics are not simple[4]: HLHS is thought to be the outcome of multiple genetic factors that interact in an environmentally sensitive way[5,6]. Likewise, large-scale studies quantifying physiological traits with continuous individual variation, such as left ventricular parameters[7], volume measures of the right heart chambers[8], trabeculation phenotypes of the left ventricle[9], and heart rate[10,11] indicate a polygenic contribution. As a result, studies of cardiac

[1]Centre for Organismal Studies (COS), Heidelberg University, Heidelberg, Germany. [2]Department of Pediatric Cardiology, Heidelberg University Hospital, Heidelberg, Germany. [3]German Centre for Cardiovascular Research (DZHK); Partner Site Heidelberg/Mannheim, Heidelberg, Germany. [4]Heidelberg Biosciences International Graduate School (HBIGS), Heidelberg University, Heidelberg, Germany. [5]European Molecular Biology Laboratory, European Bioinformatics Institute (EMBL-EBI), Cambridge, UK. [6]Cardiovascular and Metabolic Sciences, Max Delbrück Center for Molecular Medicine in the Helmholtz Association (MDC), Berlin, Germany. [7]Department of Cardiology, Heidelberg University Hospital, Heidelberg, Germany. [8]National Institute for Basic Biology, National Institutes of Natural Sciences, Okazaki, Aichi, Japan. [9]German Center for Cardiovascular Research (DZHK); Partner Site Berlin, Berlin, Germany. [10]Charité-Universitätsmedizin Berlin, Berlin, Germany. [11]Helmholtz Institute for Translational AngioCardioScience (HI-TAC) of the Max Delbrück Center for Molecular Medicine in the Helmholtz Association (MDC) at Heidelberg University, Heidelberg, Germany. [12]These authors contributed equally: Jakob Gierten, Bettina Welz, Tomas Fitzgerald. ✉e-mail: birney@ebi.ac.uk; jochen.wittbrodt@cos.uni-heidelberg.de

traits have shifted from single-gene analysis to exome- or genome-wide approaches. In CHD, most of these efforts have recovered private mutations (found only in one individual) and thus emphasize the polygenic nature of developmental cardiac phenotypes. Due to the limitations posed by a single genome, it has not been possible to assess how an individual's entire collection of genomic variants drives cardiac phenotypes in processes of health and disease. Identical twins or even better isogenic lines overcome the limitations of a single genomic context and allow incorporating the variability in individual phenotypes and genotypes, environmental factors, genetic relatedness and population stratification[12–15].

Here, we turned to a well-established vertebrate model to further our understanding of genes influencing quantitative cardiac phenotypes. We use inbred strains of the teleost medaka[16] (*Oryzias latipes*[17] and *Oryzias sakaizumii*[18]) to resolve genomic variant complexity underlying quantitative cardiac trait variability. These strains represent fixed states of individually composed natural genetic variants crossed to isogenicity. They allow for capturing a snapshot in the spectrum of phenotypic variation and establishing correlations with the underlying genotype by quantitative readouts in a controlled environment, providing a significant complementary approach to overcoming the obstacles of human studies[19]. To probe genetically determined cardiac phenotype variability, we used heart rate as a core phenotype, regulated by complex interactions of cardiac properties, including electrophysiology, morphology, and function, associated with disease loci and genetic predictors of mortality[10,11].

## Results

To get an overall picture on the genetic contribution to heart rate control we examined the phenotypic distributions of five highly inbred medaka strains using automated heartbeat detection under controlled temperature conditions. We identified two strains, HO5 and HdrR, that differ most profoundly in terms of the basal heart rate, assessed at different temperatures (21 °C, 28 °C, 35 °C). Morphological and functional analysis revealed a hypoplastic ventricle

phenotype in the HO5 strain with decreased cardiac output, which is compensated early through an increased baseline heart rate and severely impairs fitness upon reaching adulthood. Leveraging this phenotype contrast, in a cross of the extreme strains (HO5 and HdrR) we performed an F2 population analysis. We identified 59 significant loci with a maximum association peak on medaka chromosome 3 in a quantitative trait locus (QTL) mapping based on heart rate and whole-genome sequences of 1192 individually phenotyped embryos of a two-generation segregation population. We refined these loci with differential gene expression analysis in the hearts of the parental strains and teleost-to-human comparisons and validated their causal impact in vivo by CRISPR-Cas9 and base editor-mediated targeted gene editing. This demonstrated their pivotal role on heart rate, development, conduction, and morphology and revealed previously associated but functionally unrecognized loci of cardiac arrhythmia.

### Heart phenotype contrasts

The heart rate is a product of the properties of the conduction system, structural morphology, and cardiac function, and thus can be used as a readout of the way these properties are connected. We used automated microscopy and image-based heartbeat quantification under controlled physiological conditions[20], to study the maximum range of differences between phenotypes in highly isogenic inbred medaka strains/species from southern (HdrR, Cab, HO5; *O. latipes*[21]) and northern (Kaga, HNI; *O. sakaizumii*[18]) Japan (Fig. 1A). We selected inbred strains/species of different geographic origins, to maximize the genetic differences and consequently cardiac phenotypes. Heart rates stabilize over developmental time, and we addressed the dynamics of embryonic heart rates in 4-h intervals from the onset of heartbeat to the pre-hatching stage (Fig. 1B). The heart rate profiles we obtained revealed a significant and consistent spread between the five inbred strains, with the most prominent contrast between the southern *O. latipes* strains HO5 (fastest heart rate) and HdrR (slowest heart rate, Fig. 1B).

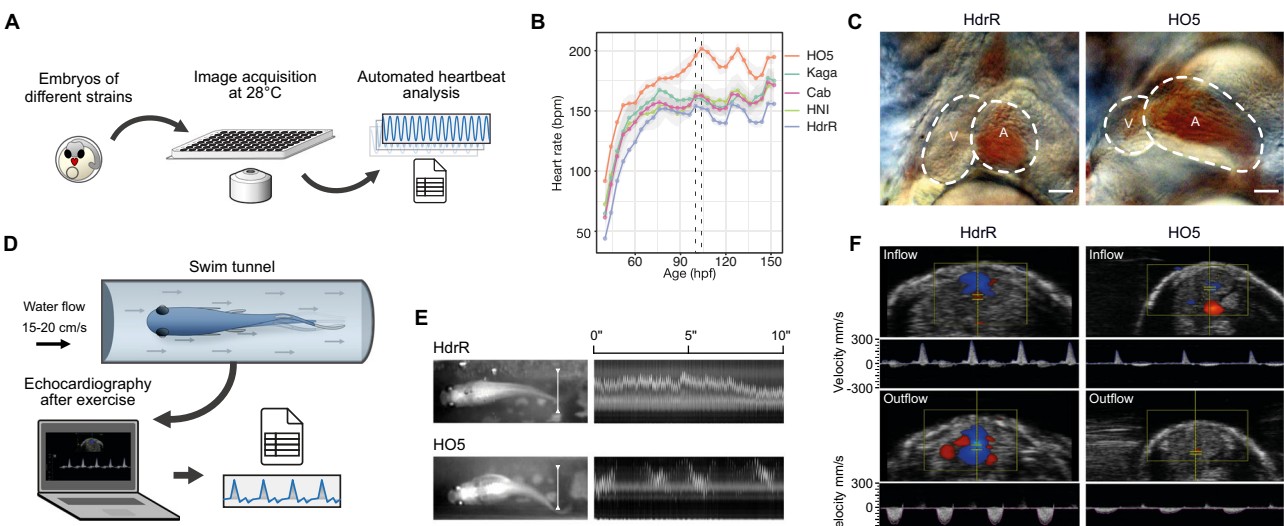

**Fig. 1 | Cardiac phenotype contrast in medaka inbred strains. A** Layout of the automated heartbeat detection in medaka embryos in native environment (28 °C) using high-throughput imaging and image-based heart rate quantification; positional effects were excluded; (**B**) Distribution of embryonic heart rates in five inbred strains derived from Southern Japanese medaka populations (HdrR, HO5, Cab) and Northern Japanese populations (HNI, Kaga) across embryonic development starting with the onset of heartbeat. Heart rates of 6–18 embryos per strain (Table S1) were determined every 4 h under a 12 h-light/12 h-dark cycle; heart rate measurements for each strain plotted as mean heart rates ± SD; dotted lines, window of circadian-rhythm-stable heartbeat for comparative analysis (100–104 h post fertilization, hpf). **C** Cardiac morphology in HdrR and HO5 hatchlings; end-systolic frame, scale bar, 50 μm (*n* = 3 for each strain). **D, E** Exercise assessment and swim performance of adult fish in a swim tunnel assay (movie S1). White line in (**E**) used for kymograph - note stable swimming behavior in HdrR versus fluctuating HO5 individual. (**F**) Pulsed-wave (pw) doppler of ventricular inflow (atrium-ventricle) and outflow (ventricle-bulbus arteriosus) tracts. Schemes are adapted from a previous publication[20] under a CC BY 4.0 Creative Commons licence.

We ensured consistency by measuring heart rates in a narrow time window between 100 and 104 hpf (between dashed lines, Fig. 1B) after the completion of critical stages of cardiac development[22] and thus avoided the impact of circadian oscillations observed from 3 days post fertilization (dpf) onwards[20].

The differences in embryonic heart rate at 4 dpf (raised at 28 °C) were substantial, prompting us to examine cardiac morphology. HO5 embryos exhibited disproportional heart chambers with an enlarged atrium and underdeveloped (hypoplastic) ventricle. This apparent hypoplastic ventricular morphology was associated with a high basal heart rate (Fig. 1C), potentially compensating for a reduced ventricular output. To address to what extent differences in embryonic heart rate are a good predictor for adult phenotypes, we next assessed the extent to which physical fitness and cardiac function are affected at adult stages. We subjected adult HO5 and HdrR individuals to a swim tunnel exercise protocol (Fig. 1E). Video monitoring indicated the efficient performance of HdrR; fish assumed stable positions in a defined water flow, requiring minimal fin excursions to generate the necessary swimming speed (Fig. 1F and movie S1). In contrast, HO5 individuals (apparently higher BMI) used almost the entire body length to generate forward movement. In addition, they failed to assume stable positions within the constant water flow (cf. kymograph in Fig. 1E). Histological assessment of the skeletal muscle structure showed regular organization in adult HO5 with no apparent difference to HdrR (Fig. S1), which does not support a skeletal muscle phenotype as a primary cause of reduced physical performance. Following swim tunnel exercise, we found that HO5 exhibited significantly reduced velocities in intracardiac blood flow, indicating reduced myocardial function, as observed using echocardiography, including pulse-waved (pw) Doppler measurements of the ventricle in- and outflow (Fig. 1F). These findings argue for early alterations of physiological cardiac traits that occur in HO5 embryos and severely impact on physical fitness and cardiovascular health in adulthood.

## Segregation analysis

To map the loci contributing to these complex phenotypes, we established an F2 mapping population derived from the two extreme parental strains (HO5 and HdrR) in which we correlated heart rates and whole-genome sequences (WGS). To model a human-relevant modifiable environmental factor of embryonic development, we turned to the well-established heart rate increase in elevated water temperatures[20] and measured embryonic heart rates of the parental strains HO5 and HdrR, as well as in F1 and in the F2 mapping population at an incremental ambient temperature ramp (21 °C, 28 °C, 35 °C). For mapping we employed a two-generation segregation design and used single-nucleotide polymorphisms (SNPs) as markers (Fig. 2A). SNP calling in HO5 WGS against the HdrR reference genome established 979,713 differential homozygous SNPs as marker system.

We crossed the two isogenic strains HO5 and HdrR, with distinctly differentiated cardiac phenotypes, to generate a hybrid F1 population with haploid sets of chromosomes from each parental strain and observed heart rates that were intermediate between the two parental strains. We next performed F1 intercrosses and sampled 1260 individuals of the F2 generation with unique recombined genotypes (linkage blocks) to serve as the mapping population (Fig. 2A). We conducted individual heart rate phenotyping of all 1260 F2 embryos and found (with a few outliers showing sporadic arrhythmia) that the phenotypic distribution covered the range between the parental phenotypes at all three temperatures (Fig. 2B). This reflects the differential segregation of multiple alleles that directly impact the heart rate or secondarily due to morphological-functional effects on the heart.

## Genome-wide QTL mapping

We raised 1192 individually phenotyped F2 embryos in 96-well plates, followed by genomic DNA extraction, and whole-genome-sequencing.

We sequenced 1192 samples with an average coverage of 0.78×. We used a three-state Hidden Markov Model (HMM) to segment all crossover locations and determine genotype states (AA, AB, BB) based on SNPs homozygous divergent in HO5 (AA) and HdrR (BB). Interestingly, we observed a distortion in the expected Mendelian ratio of 1:2:1 (AA:AB:BB alleles) with an overrepresentation of AA (HO5) alleles, suggestive of a potential reference bias. This bias was evident in most F2 offspring samples (Fig. S2) and was not restricted to specific regions of the genome. Although it could be possible that certain crossover events between HdrR and HO5 are incompatible, the most parsimonious explanation is a tendency towards homozygous reference calls within the SNP genotype calls used to train the HMM. Having called crossover events and generated a recombination map across all F2 offspring samples independently, we merged the crossover locations and segmented the genome at every breakpoint, resulting in a genotype matrix containing AA:AB:BB calls for variable-sized blocks. The median genotype block size, once segmented across all F2 cross-recombination positions, was 24 kb.

Using this genome matrix we then conducted genome-wide association analyses of 101,265 segmented recombination block regions on individual heart rate measures at three different ambient temperatures and for absolute differences in repeated measurements across all temperatures 21 °C, 28 °C and 35 °C (variance phenotype), using a linear mixed model from the GridLMM software package[23]. We found significantly associated QTLs that were mostly consistent across the three temperatures (Fig. 2C) and different peaks in the test on heart rate variance from 21-35 °C (Fig. S3). Overall, we detected 1385 significant loci across all phenotype tests and performed collapsing down to 59 distinct fine-mapped regions after linkage disequilibrium (LD)-based SNP pruning. The maximum achievable resolution for loci fine mapping was 10 kb due to the window size used for recombination block mapping, and the median size across all fine-mapped loci was 115 kb. We were able to fine-map to 10 kb for only 5/59 loci; however, most fine-mapped regions contained small numbers of genes (median of 1 and a range between 0 and 36 genes per fine-mapped block). Overall, we detected similar numbers of fine-mapped loci for the different phenotype measures we tested, with 17, 16, 8, and 17 unique fine-mapped regions for 21 °C, 28 °C, 35 °C, and the variance-based phenotype, respectively.

We prioritized further candidate gene analysis based on their expression in the adult heart. We sequenced total RNA of heart tissues in 4 samples from both the HO5 and HdrR strains. After quality control 18,321 genes (75% of known medaka genes) had sufficient coverage to allow differential expression analysis. At a false discovery rate (FDR) of 0.01 and minimum fold change of 2, we detected 1161 significantly differentially expressed genes between HO5 and HdrR hearts (Fig. S4A), which we then used to prioritize genes within significantly associated F2 recombination blocks for subsequent validation experiments. Genes expressed in liver samples of both strains were used as baseline reference (Fig. S4B). Additionally, we assessed the likely impact of SNP calls in HO5 against the HdrR reference using the variant effect predictor (VEP) from Ensembl[24]. As expected, loss-of-function (LoF) variants were rare within these blocks, with a median of 0 variants and maximum of 3 per gene. However, using the combination of information, it was possible to rank genes and variants more likely to be impactful for the observed phenotypic differences. One case supported by LoF variation is a single premature stop codon variant in the *blzf1* gene, and in contrast, the *rrad* gene, where no LoF variants in HO5 were observed but a significant differential expression in heart tissues.

## Gene enrichments

Across all 59 significantly associated fine mapped regions there were 173 annotated genes in the *Oryzias latipes* (HdrR) ENSEMBL reference; 28 of the fine mapped regions had no annotated gene within

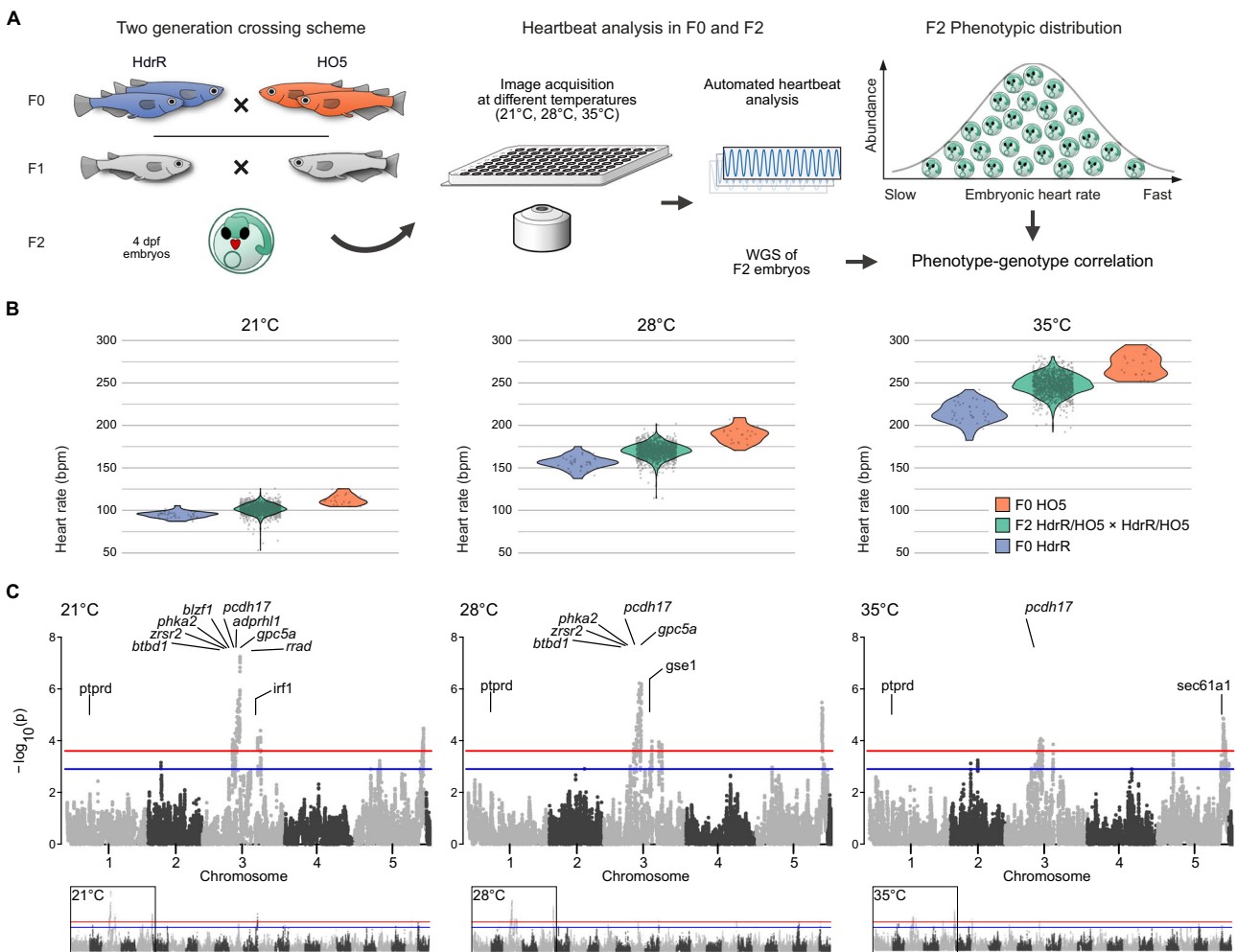

**Fig. 2 | F2 segregation analysis reveals temperature-sensitive QTLs affecting the heartbeat. A** Crossing setup used to generate HdrR × HO5 offspring with segregated SNPs in the second generation: isogenic HO5 and HdrR parents are crossed to generate hybrid (heterozygous) F1 generation (gray) with intermediate phenotype, which after incrossing results in F2 individuals (green) with individually segregated SNPs resulting from one cycle of meiotic recombination. Automated embryonic heartbeat analysis at 4 days post-fertilization (dpf), performed on all 1260 individual embryos and challenged by increasing temperatures (21 °C, 28 °C, 35 °C). Whole-genome sequencing of measured individuals (1192 genomes) with an effective average coverage of 0.78× allows phenotype-genotype correlation. **B** Individual embryonic heart rates of the inbred strains HdrR (slow heart rate, blue) and HO5 (fast heart rate, orange) increase with temperature (21 °C, 28 °C, and

35 °C). The F2 individuals with recombined HdrR × HO5 genomes (green) span the range of parental (F0) heart rates between the two strains with a subgroup of F2 individuals exhibiting heart rate variance beyond the parental extremes (21 °C and 28 °C); sample sizes (*n*) for 21 °C, 28 °C and 35 °C: *n* (HdrR F0) = 35, 35, 35, n (HO5 F0) = 20, 22, 22, n (HdrR × HO5 F2) = 1260, 1260, 1260. **C** Minus log10 *p* values from genome-wide association tests of recombination block genotypes and heart rate measures at different temperatures using a linear mixed model. Chromosomes 3 and 5 hold the most segregated recombination blocks associated with heart rate differences. Twelve selected genes from the loci passing the significance threshold are indicated; red line: 1% false discovery rate (FDR); blue line: 5% FDR, determined by permutation. Schemes are adapted from a previous publication[20] under a CC BY 4.0 Creative Commons licence.

their boundaries (8, 10, 6 and 4 loci with no annotated genes for heart rate phenotypes at 21 °C, 28 °C, 35 °C, and variance respectively), leaving 31 containing 1 or more genes with a median of 3 per fine mapped region. The largest region was found on chromosome 1, associated with the variance phenotype and contained 36 genes, only two of which had any LoF variant called in HO5 against the HdrR reference (Data S1, block ID-2). Using the 173 annotated genes, we performed multiple gene enrichment models using the PANTHER overrepresentation test (released 20230705), gProfiler functional profiling, and functional annotation clustering using DAVID[25–27]. Using the overrepresentation test from PANTHER, we found significant associations (FDR *P* < 0.05) for 72 different biological process GO terms, including "thrombocyte differentiation", which has a greater than 100-fold enrichment. Both gProfiler and DAVID found a significant enrichment (adjusted *P* value 4.393 × 10-10) for the GO term "galactoside binding" with 6 of the 172 genes being involved in the binding of glycoside carbohydrate derivatives, as well as a weaker but

significant enrichment (adjusted *P* value 1.895 × 10-2) for the KEGG term "Linoleic acid metabolism". For functional annotation clustering using DAVID with a classification stringency set to Medium using the Benjamini adjustment, there were 21 clusters defined, the strongest of which had an overall enrichment score of 4.56 and included the GO terms "galactoside binding" and "laminin binding".

Recent studies in humans have looked at the pleiotropic regulatory activities of Galectins in relation to cardiovascular disease (CVD) and their proinflammatory role in the atherosclerotic and plaque formation process[28]. Here we found 6 Galectin-related genes (ENSORLG00000030479, ENSORLG00000026315, ENSORLG00000010700, ENSORLG00000010715, ENSORLG000000-10697, ENSORLG00000024256) to be associated to heart rate differences in medaka, associated to the variance in heart rate, and found within the same fine-mapped region at chromosome 8:13655000-13935000 (Data S1, block ID-42). Galectins are promiscuous, with multiple cellular functions and impede arteriogenesis[29]. Consequently,

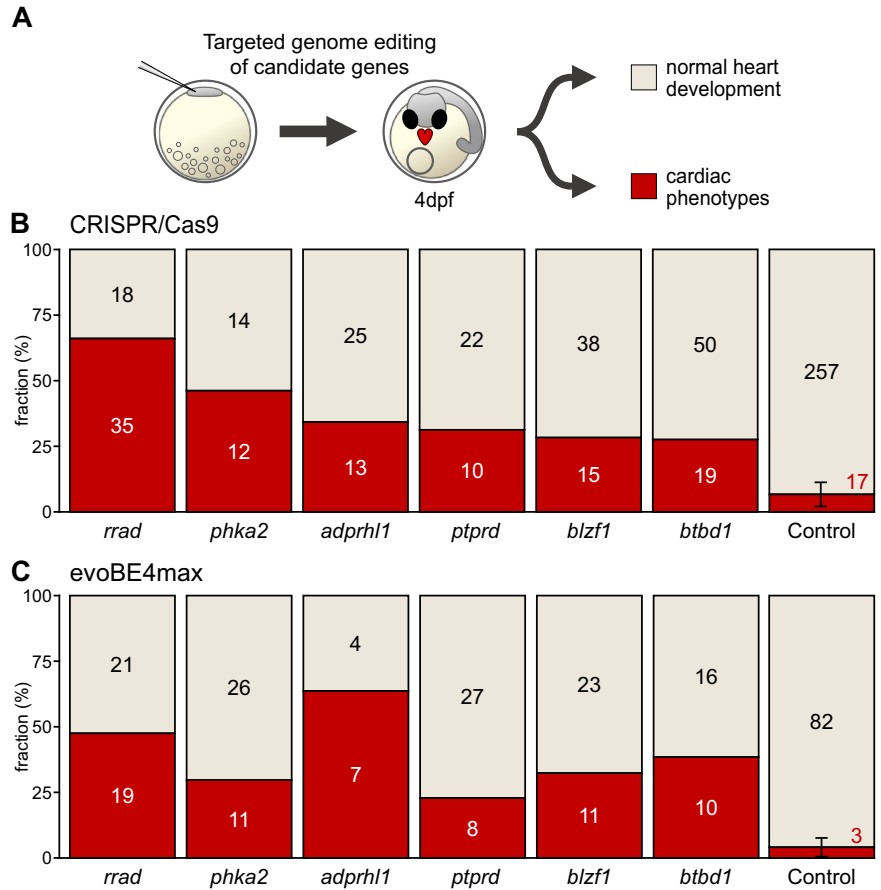

**Fig. 3 | Phenotype proportions in knockout models of cardiac candidate genes.** **A** Validation workflow encompassing zygotic microinjections using a HdrR (*myl7::eGFP*; *myl7::H2A-mCherry*) reporter line, followed by phenotypic classification of embryos with normal developed hearts and embryos with cardiac-specific phenotypes 4 days post-fertilization (dpf). **B** Proportion (bars) and counts (values) of cardiac-affected and normally developed embryos after CRISPR-Cas9-mediated knockout of indicated candidate genes versus control (mock injection). Phenotypic proportions of crispants were determined from 54 to 100 embryos and compared to 303 mock-injected control embryos. The mean and standard deviation of the control group were calculated based on seven technical replicates. **C** Independent replication using base editing: phenotypic distribution (proportion/bars and counts/values) resulting from targeted gene editing mediated by introducing premature termination codons via the cytosine base editor evoBE4max and a set of distinct guide RNAs targeting the same genes as in (**B**). Phenotypic proportions of editants were determined from 50 to 75 embryos and compared to 105 mock-injected control embryos. The mean and standard deviation of the control group were calculated based on three technical replicates.

galectin inhibitors are an attractive target for therapeutics pertaining to remodeling in myocardial infarction[30]. Thus, associations based on medaka heartbeat dynamics directly revealed genes with highly relevant translational implications.

## In vivo validation

Our genome-wide QTL mapping based on heart rate metrics identified loci associated with biological functions in heart development, cardiac function, and electrophysiology. To provide evidence for a causal role of the identified loci, we selectively edited genes in medaka embryos and investigated the respective impact on embryonic heart rate and morphology in vivo. After analyzing the search space (Table S2), we narrowed our selection to 12 candidate genes based on their high linkage probabilities, robust cardiac expression and differential expression in medaka heart, novelty, and evolutionary conservation/human relevance (cf. Methods "Candidate gene selection"): *adprhl1*, *blzf1*, *btbd1*, *gpc5a*, *gse1*, *irf1*, *pcdh17*, *phka2*, *ptprd*, *rrad*, *sec61a1*, and *zrsr2*, where *rrad* and *adprhl1* have previously been directly associated with cardiovascular traits.

We addressed potential functional effects on the heart following targeted editing of candidate genes by CRISPR-Cas9 or base editors. Phenotype analysis was performed after cardiovascular development was expected to be complete at 4 dpf (28 °C) in embryos devoid of global developmental defects, which resulted in two major phenotypes: embryos with morphologically normally developed hearts and cardiac-specific affected embryos, showing looping defects, pericardial edema, arrhythmia, or aberrant atrial or ventricular size and morphology (Figs. 3 and S5, Table S3; Movies S2 and S3). Cas9-based targeted editing of candidate genes prominently increased the proportion of cardiac phenotypes above the baseline threshold established by injected controls (max. 7% heart phenotypes; Figs. 3B and S5). Specifically, *adprhl1*, *blzf1*, *btbd1*, *phka2*, *ptprd*, and *rrad* showed high rates of prominent heart phenotypes in the CRISPR-Cas9 approach. To underline the specificity of observed phenotypes, we corroborated the results of the CRISPR-Cas9 gene interrogations in an independent experimental series introducing premature termination codons (PTCs) with the cytosine base editor evoBE4max for six of the twelve selected candidate genes (Fig. 3C).

To assess the primary effects of edited genes on heart rate, we profiled heart rates in successfully injected (Cas9 or base editor) embryos at three different temperatures. In the two groups with apparently normal heart development or cardiac phenotypes (cf. Fig. 3), we found a higher number of genes showing significantly altered baseline heart rates at the test temperatures and across gene editing methods in cardiac affected embryos, likely reflecting secondary effects of aberrant morphology and function on heart rate

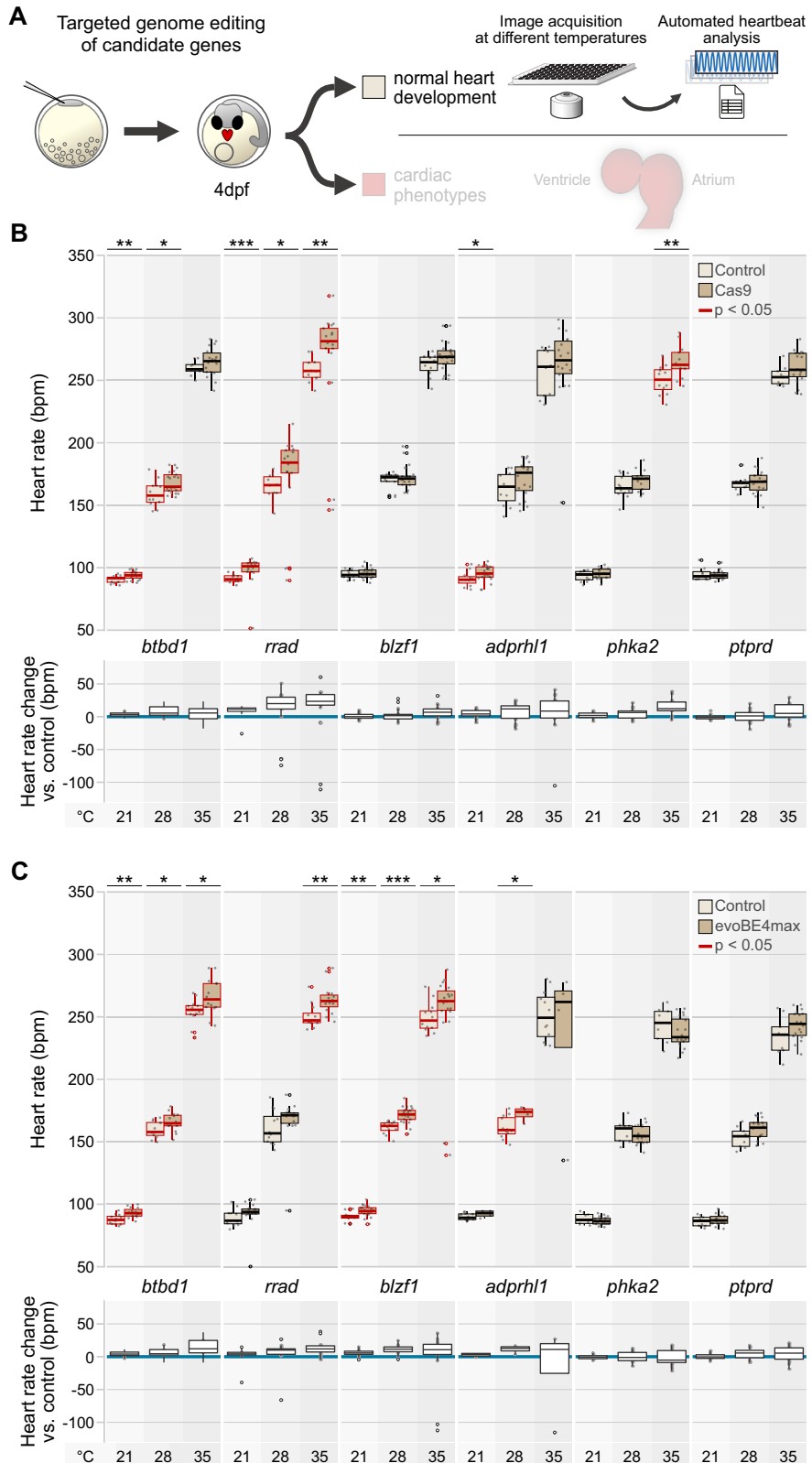

(Fig. S6). Thus, to exclude secondary effects in embryos with aberrant cardiac morphogenesis, we applied the automated heart rate assay in embryos with apparently normal heart development (Figs. 4 and S7). We found that for eight out of the twelve candidate genes, the heart rate was affected (Data S2), while six of them showed an increase in heart rate at the three different temperatures tested in replicates with Cas9- and base editor evoBE4max-mediated targeted gene editing

(Fig. 4B, C). Interfering with *zrsr2* and *gse1*, conversely, resulted in a marked decrease in heart rate (Fig. S7).

Our automated heartbeat quantification algorithm is optimized to extract heartbeats from cardiac segments with the most robust image signal, not discriminating between the atrial and ventricular frequencies. Consequently, this heartbeat quantification uncovers arrhythmias as apparent as marked outliers in the case of *rrad*, *adprhl1*,

**Fig. 4 | Editing of candidate genes affects embryonic heart rate in medaka embryos. A** Validation workflow including zygotic microinjections using a HdrR (*myl7::eGFP*; *myl7::H2A-mCherry*) reporter line, high-throughput image-based heart rate quantification in normally developed injected specimens at 4 days post fertilization (dpf). **(B, C)** Heart rate distributions and absolute heart rate differences of morphologically normal crispants (*n* = 13 to 22 embryos) and editants (*n* = 4 to 22 embryos) compared to mock-injected control embryos (*n* = 8 to 12 embryos) at 21 °C, 28 °C, and 35 °C with CRISPR-Cas9- (B) and base editor-mediated (**C**) targeted mutagenesis of candidate genes (heart rate values are listed in Data S2 and sample

numbers in Data S3). The significance of heart rate differences between the edited group and its corresponding control was assessed with the two-sided Wilcoxon test; *, *p* < 0.05; **, *p* < 0.01; ***, *p* < 0.001 (*p* values listed in Data S4). Data is visualized as box plots (median+/− interquartile range between the 25th and 75th percentiles) and overlaid scatter plots of heart rate measurements; knockouts of *adprhl1*, *btbd1*, *blzf1*, *phka2*, and *rrad* have temperature-dependent effects on heart rate. Schemes are adapted from a previous publication[20] under a CC BY 4.0 Creative Commons licence.

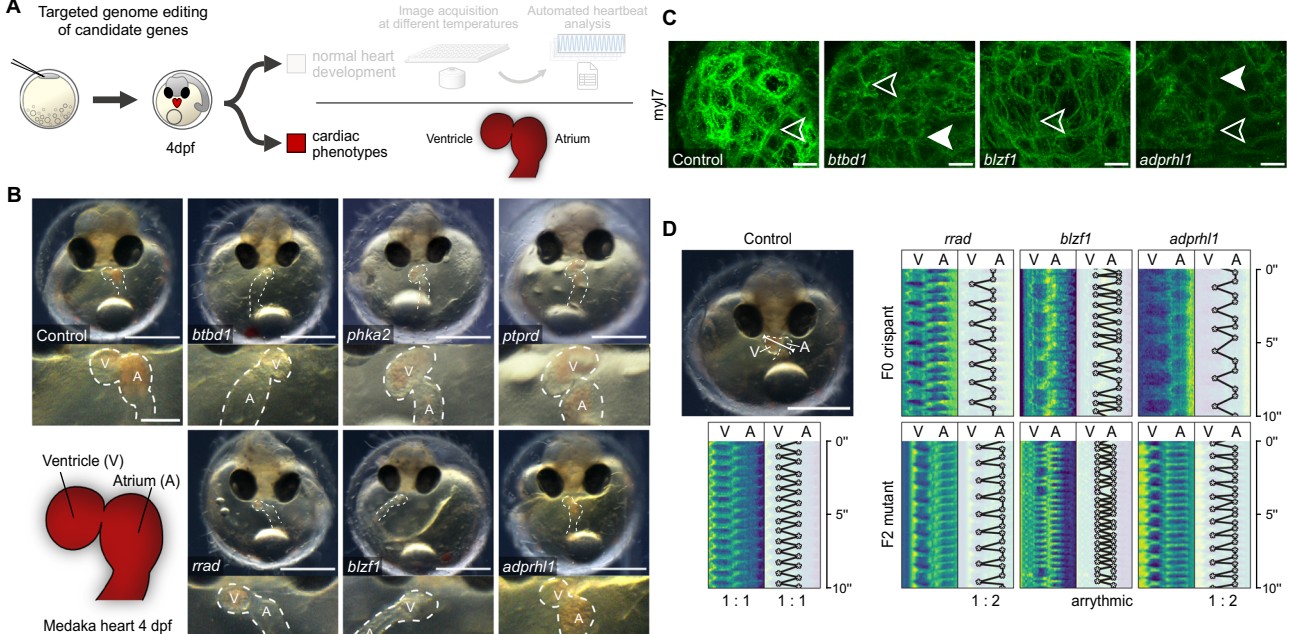

**Fig. 5 | Developmental and electrophysiological phenotypes in F0 crispants and F2 mutants. A** Workflow including zygotic microinjections using an HdrR (*myl7::eGFP*; *myl7::H2A-mCherry*) reporter line and phenotyping of cardiac-affected crispants 4 days post-fertilization (dpf). **B** Representative cardiac phenotypes of knockout embryos for all six candidate genes targeted with CRISPR-Cas9 and base editor compared to a control embryo at 4 dpf; bright-field overview of the injected specimen (top; scale bar, 500 μm), close-up image of the heart (bottom; scale bar, 125 μm); cf. movie S2. **C** Immunostaining of regulatory myosin light chain (myl7) in crispants of *btbd1* (*n* = 5 embryos), *blzf1* (*n* = 6 embryos), and *adprhl1* (*n* = 7 embryos) at 4 days post-fertilization (4dpf) in Cab. In *btbd1* and *adprhl1* mosaic knockout, the

myosin signal is reduced and sarcomeric structure disturbed (filled arrowhead) compared to *blzf1*, which shows no differences in contrast to control (*n* = 2 *embryos*) (unfilled arrowhead); (scale bar, 10 μm). **D** Heart rhythm analysis in F0 crispants and F2 mutants as depicted by kymographs derived from atrium (A) to ventricle (V) spanning line selection in 10 s time-lapse movies (scale bar, 500 μm). In contrast to the regular rhythm in the control embryo (left), the representative *rrad*, *blzf1, and adprhl1* F0 crispant and F2 mutants (right) display different degrees of atrioventricular block; cf. movie S4. Schemes are adapted from a previous publication[20] under a CC BY 4.0 Creative Commons licence.

and *blzf1* (Fig. 4B, C). Manual examination of the corresponding image series revealed embryos with aberrant atrioventricular conduction and block, resulting in a marked drop in the automatically scored ventricle contraction rate.

We further closely assessed morphological aberrations and functional cardiac-specific phenotypes (quantified in Fig. 3) and the gene editing cases triggering arrhythmia (observed in Fig. 4B, C) (Figs. 5 and S8). While targeted editing of *ptprd*, *blzf1*, and *btbd1* distorted cardiac looping (incorrect positioning of the atrium to the right and the ventricle to the left), a reduced ventricle size was observed in specimens after targeted editing of *rrad*, *adprhl1*, *blzf1*, or *btbd1* (Fig. 5B). We further analyzed cardiac sarcomere structure in three candidate genes, *btbd1*, *blzf1*, and *adprhl1*, which were selected based on cardiac expression data and observed cardiac phenotypes related to cardiac looping, ventricular size, and conduction phenotype. Immunostaining of a regulatory myosin light chain component (myl7 antibodies) of thick filaments was used to assess the sarcomeric structure in crispants of respective candidate genes at 4 dpf. Among the tested candidates, editing of *btbd1* and *adprhl1* led to prominently reduced myosin light chain signal and

sarcomeric disarray compared to mock-injected wild-type embryos (Fig. 5C). This finding is in line with the previously proposed role of *adprhl1* in myofibril assembly in *Xenopus*[31]. Heartbeat analysis revealed cardiac conduction defects in crispants of three candidate genes (*rrad*, *blzf1*, *adprhl1*; Fig. 4B, C). Interestingly, these genes are contained within multi-gene loci previously associated with electrophysiological phenotypes and disorders: *blzf1* is contained in a locus associated with cardiac repolarization (QT interval)[32], *rrad* located in a locus associated with the specific human cardiac conduction disorder Brugada syndrome[33], and a locus containing *adprhl1* with left anterior fascicular block[34]. Strikingly, none of these associations had been experimentally validated. We re-assessed the phenotype in compound heterozygous F2 embryos and verified the initially observed AV block with 2:1 conduction in the case of *rrad* and *adprhl1*, while *blzf1* mutants displayed an irregular lack of AV conduction (Fig. 5D).

Taken together, the F2 gene segregation workflow combined with quantitative phenotyping and targeted mutagenesis for validation allowed to robustly uncover novel genetic risk factors for human heart disease.

## Discussion

Using apparent heart rate differences in embryos of highly inbred medaka strains, we mapped specific SNPs that contribute to differences in heart rate, rhythm, and ventricular size disproportion, phenotypes interrelated in human CHD. Analyzing almost 1200 genomes and the corresponding heart rates as readout with predictive power[10,11], we identified 59 QTLs at high resolution. 173 of the genes contained in these mapped loci showed a direct link to heart as well as extracardiac organ functions, reflecting the polygenic architecture of quantitative heart phenotypes.

Considering expression in the medaka heart and an orthology-guided examination of genes associated with human heart phenotypes, we focused on prominent candidates that we specifically edited in their genomic context to address and validate their potential role in the development of cardiac phenotypes. CRISPR-Cas9 and base-editing-mediated targeting of these candidates were carried out, and uncovered genes involved in heart rate and rhythm control as well as structural development of the heart. Intriguingly, using heart rate metrics for phenotype-genotype mapping, we identified genes with dual roles affecting heart muscle mass and rhythm; e.g., *rrad* combining regulative functions on cardiac muscle strength and electrophysiology. *Rrad* is known to inhibit cardiac hypertrophy through the CaMKII pathway with implications for heart failure[35], and is associated with hypertrophic cardiomyopathy (HCM) phenotype in RRAD-deficient cell line[36]. *Rrad* mutations detected in a specific form of familiar arrhythmia (Brugada syndrome) trigger cytoskeleton and electrophysiological abnormalities in iPSC-CMs[33]. Of note, a subset of Brugada syndrome patients is susceptible to developing high-risk atrioventricular block with direct implications for the mode of implantable defibrillator and cardiac pacing therapy[37]. Given that the genetics of Brugada syndrome patients with high-risk atrioventricular block remains elusive, our genetic mapping and in vivo Cas9- and base-editing-mediated knockout models identifying *rrad* variants to induce a specific conduction phenotype with high-penetrance atrioventricular block, add insights on *RRAD*'s functional significance in humans and could have immediate value for antiarrhythmic therapy and device selection.

Additionally, we found and validated the heart rate-associated genes *blzf1*, located on a locus associated with cardiac repolarization but previously not recognized as a functional candidate[32], and *adprhl1*, associated with a ventricular conduction disorder[34], resulting in specific conduction defects in our generated knockout models. This demonstrates the power of complex cardiac trait analysis in medaka inbred strains to detect and validate human-relevant association signals, guiding genome editing experiments to uncover novel players and potential targets in heart disease.

We noticed a significant decrease in ventricular size in a subset of gene knockouts (*rrad*, *adprhl1*, *blzf1*, and *btbd1*), and it appears that *adprhl1* has a major impact on ventricular dimensions. Adprhl1 is a conserved pseudoenzyme with ADP-ribosylarginine hydrolase and magnesium ion binding activity, likely exclusive to the heart. Transcriptomics revealed very strong cardiac expression (3109.244 RPM compared to weak (1.175 RPM) in the liver) in medaka. Until now, *adprhl1* has been studied functionally only scarcely. Of note, in *Xenopus*, *adprhl1* can localize to the cardiac sarcomeres. While morpholino-mediated knockdown of *adprhl1* did not affect early cardiogenesis, it disrupts myofibril assembly in the forming ventricle and leads to small, inert ventricles[31]. Targeted gene editing (Cas9) and cardiac immunostaining of myosin filaments in medaka embryos showed disorganized sarcomeric patterns in *adprhl1* crispants, further supporting its suggested role in myofibrillar assembly. In light of the hypoplastic ventricle observed in the HO5 strain, its location on a strongly associated QTL on chromosome 3 in medaka, and targeted gene editing in vivo, we provide evidence for *adprhl1* as a strong contender for controlling myofibril assembly and ventricular outgrowth.

The homozygous fixation of causative genomic variants in viable inbred medaka strains allows not only for modeling twin studies with arbitrary scalability but also for longitudinal investigation to estimate the variants' effects within a lifespan. Notably, we found an inverse relation of metrics for embryonic heart rate and ventricle size in the HO5 strain, which exposes a pathological heart phenotype with impaired cardiac function and physical performance in exercise tests, demonstrating the predictive power of genotype-specific embryonic heart rate profiles. Finding such early biosignatures is essential for diagnosing and potentially preventing severe patho-physiologies that become less likely to be correctable over the long term.

Most cardiac disease phenotypes are quantitative and manifest in a health-disease continuum. Here, we demonstrate that inbred medaka represents a powerful resource to partition and validate the spectrum of phenotypes in a strain-specific way. Examining the extremes of a physiological trait range bears the power to reveal the genetic factors likely associated with disease susceptibility. The F2 cross of inbred strains with contrasting phenotypes allows to segregate causal alleles in a way that cannot be achieved with human genome-wide association studies (GWAS)[19,38]. Nevertheless, the results can be immediately transferred to the human context and shed light on genetic variations that have flown under the radar.

In summary, our research underscores the invaluable role of inbred medaka strains in unraveling the intricacies of cardiac development, physiology, and pathology. The comprehensive genetic analysis conducted here has not only led to the discovery of novel heart-related genes but also established a promising, robust framework for future investigations.

By leveraging the power of a scalable medaka inbred panel[19,39,40] in conjunction with highly reliable quantitative phenotyping assays[41,42] across the entire phenotypic spectrum, we anticipate a crucial enhancement in our ability to pinpoint and validate combinatorial candidate variants across both coding and non-coding regions of the genome. This holds a great perspective for advancing our understanding of cardiac biology in particular and may pave the way for innovative therapeutic interventions in CVD.

## Methods

### Fish maintenance

The wild-type Kaga, HNI, Cab, HO5, and HdrR strain and a fluorescent cardiac reporter line HdrR (*myl7::EGFP myl7::H2A-mCherry*) were used in this study. All fish stocks were maintained (fish husbandry, permit number 35−9185.64/BH Wittbrodt) and experiments (permit numbers 35−9185.81/G-145/15 and 35−9185.81/G-271/20) were performed in accordance with local animal welfare standards (Tierschutzgesetz §11, Abs. 1, Nr. 1) and with European Union animal welfare guidelines[43]. Fish were kept as described previously in ref. 44. The fish facility is under the supervision of the local representative of the animal welfare agency.

### Automated microscopy and heartbeat detection of medaka embryos

The automated microscopy and heart rate quantification were applied as described previously[20]. Embryonic heart rate profiles over developmental stages were generated by imaging embryos in 96-well plates every 4 h under a 12 h-light-12 h-dark-cycle, i.e., 12 h dark minus imaging time of 20 min/plate/time point; incubation temperature was 28 °C (developmental profiles) and 21 °C, 28 °C, and 35 °C in the screen.

All F0, F1, and F2 embryos were imaged as separate batches in 96-well plates each with one row (12 embryos) of Cab embryos as an internal control. For phenotype-genotype correlations in individual F2 embryos, fifteen 96-well plates, each containing two F2 crosses (HdrRf × HO5m F2 and HO5f × HdrRm F2), were imaged and processed under the same conditions.

To exclude positional effects of plate coordinates on heart rate, a full 96-well plate with Cab embryos was recorded at 28 °C. Heart rates

were normally distributed ($P = 0.86$; Shapiro–Wilk normality test). One-way ANOVA indicated no significant effects of plate row on mean heart rates (degrees of freedom (df) = 7.88; $P = 0.61$), nor of plate column on mean heart rates (df = 11.84; $P = 0.15$).

## Swim tunnel assay and echocardiography of adult medaka fish

Two months before the experiments ("Kardiale Phänotypisierung adulter Medaka Fische". Aktenzeichen: 35-9185.81/G-10/17), HdrR F96 and HO5 F112 were kept at 22 °C water temperature and 14 h/10 h light/dark conditions. Swim tunnel (Loligo Systems) assay was performed with the following flow velocity and settings. Equilibration at 3 cm/s for 20 min, then increase of water flow by 5 cm/s every 5 min; mO2 was scored at each step. Two 10 s videos with 100 fps were recorded at 15 cm/s and 20 cm/s (if applicable) during the interval after a flush. The test was stopped when fish remained 3 s or more at the rear of the chamber. Echocardiography was performed as previously described[45] with 150 mg/l tricaine and 8 mg/l metomidate.

## HdrR × HO5 intercross and phenotyping of F0, F1, and F2 embryos

For heart rate measurements in F0 (i.e., HO5 F112, HdrR F95, Cab F68), embryos were collected from these crosses: 3 female × 1 male HO5 F111, 3 female × 1 male HdrR F94, and as plate control 3 female × 1 male Cab F67.

Phenotyping F1 hybrid embryos and generation of F1 hybrid stocks: 3 HO5 F111 females × 1 HdrR F94 male resulting in stock HO5 F111f × HdrR-II F94m F1, and 3 HdrR F94 × 1 HO5 F111m resulting in stock HdrR F94f × HO5 F111m F1. Cab F68 embryos were used as plate control, obtained from 3 female × 1 male Cab F67.

The embryonic F2 screen (heart rate) was conducted within two months. Two F2 populations were derived from two separate crosses of each F1 hybrid stock (1) HO5 F111f × HdrR F94m F1 and (2) HdrR F94f × HO5 F111m F1: F2 embryos were collected from stocks (1) and (2), for each of which 3 tanks with 3 females × 1 male were set up (both F2 populations derived each from 12 fish). Cab F68 embryos were used as plate control, obtained from 3 female × 1 male Cab F67.

Incubation conditions for F0, F1 embryos: medaka embryos were raised at 28 °C with either max. 20 embryos/20 ml medaka hatching medium in 60 mm dishes or with max. 50 embryos/50 ml medaka hatching medium in 90 mm dishes.

Incubation conditions for F2 embryos: 55 embryos were cultured at 28 °C in 90 mm dishes with medaka hatching medium; Cab controls: 25 embryos in 9 cm dishes with hatching medium.

Embryo preparation for imaging: Embryos were rolled on sandpaper the afternoon/evening before imaging and placed back to the primary culture dishes with fresh hatching medium.

Heart rate assay (cf. "Automated microscopy and heartbeat detection of medaka embryos"): On the day of imaging, embryos were transferred from hatching medium to ERM before mounting. Individual F2 embryos were loaded with 150 µl ERM in 96-well plates (U-bottom). Approximate imaging times were: Start equilibration at 12:00, imaging 21 °C at 12:15 P.M., 28 °C at 13:15 P.M., and 35 °C at 14:15 P.M.

## F2 sample preparation and whole-genome sequencing

After imaging of F2 embryos in 96-well plates, ERM was exchanged with medaka hatching medium by pipetting out 140 µl ERM and adding 200 µl medaka hatching medium. Subsequently, 190 µl of the medaka hatching medium was exchanged daily until first embryos hatched. When the majority of embryos had hatched, the remaining unhatched embryos were manually dechorionated. All hatchlings were transferred from the 96-well imaging plate to identical coordinates of a 96-DeepWell plate, which was covered with an adhesive aluminum foil (4titude) and frozen acutely at −80 °C until further processing. Samples were processed at Wellcome Sanger Institute (UK).

Library preparation and WGS on a HiSeq X Ten instrument (Illumina) with an average sequencing depth of 0.78×. Library preparation was performed following the standard PCR-free Illumina protocol[46]. In total, 1 µg was picked from DNA extraction plates within the Sanger sample logistics facility and passed into the Sanger sequencing pipeline from library preparation. Following successful preparation of sequencing libraries, the samples were QCed using Qubit, and samples passing the facility quality control threshold were multiplexed sequenced in paired-end mode with 150 samples per Illumina X10 flow cell.

## Whole genome-sequence analysis and mapping

For the genotyping of recombination blocks in the F2 population, a modified algorithm of the previous version was used (https://github.com/tf2/ViteRbi). The analysis was based on SNPs homozygous divergent in HO5 (AA) and HdrR (BB). Homozygous divergent SNPs were determined by the alignment of high-depth HO5 genome sequencing data against the HdrR-II reference genome. At each homozygous divergent SNP-locus read counts supporting each genotype (A, T, C, G) were extracted from the genome alignments. In a fixed window of 5000 bp, read counts were summed and transformed into proportions of reads supporting the AA genotype. This genotype proportion was the observational input for a three-state Hidden Markov Model employing the Viterbi algorithm to segment all crossover locations and to determine genotype states (AA, AB, BB) in the resulting recombination blocks across all samples. Genome-wide association tests using recombination block genotypes and heart rate measures were performed using a linear mixed model[23].

## RNA sequencing and transcriptome analysis

Heart and liver samples were dissected from euthanized adult HO5 F112 and HdrR F96 female fish and collected into Qiagen Collection Microtubes (racked, 19560) with cabs (19566), acutely frozen on dry ice, and then stored at −80 °C. Total RNA purification was performed with QIAsymphony RNA, purifying liver samples with RNA CT 400 and heart samples with RNA FT 400. Samples were prepared for Illumina RNA sequencing using the NEBNext Ultra II Directional RNA Library Prep Kit for Illumina and sequenced on a HiSeq 4000 sequencing platform following the manufacturer's instructions. All pair-end reads were cleaned up using Fastp v0.20.0[47], aligned to the medaka HdrR reference genome Ensembl Release 98 (ASM223467v1) using STAR v2.7.3a[48], and estimated counts per gene were obtained. Differential expression analysis was performed using DESEQ2[49] with an FDR cut-off of 0.01 and fold change of 2, 1161 and 1803 significantly differentially expressed genes were obtained for heart and liver samples, respectively.

## Candidate gene selection

QTLs were prioritized according to their strength of linkage. To filter the search space defined in Data S1, we included recombination blocks (QTLs) with SNP-based genotypes significantly (log10 $p$ value > 3) associated with heart rate variation. Only the strongest linkage to heart rate variation is given for each block compared at three temperatures (21, 28, and 35 °C). Phenotype associations of orthologous human genes supported candidate gene selection relevant to human cardiac disease. Therefore, the human GWAS catalog (gwas_catalog_v1.0.2-associations_e104_r2021-10-22.tsv downloaded from https://www.ebi.ac.uk/gwas/docs/file-downloads)[50] was scanned for terms indicative for heart rate and morphology associations. Candidate genes were additionally studied with the GeneALaCart batch query processor of the GeneCards suite (https://www.genecards.org)[51]. If blocks contain multiple genes, candidates were chosen according to defined criteria, primarily novelty and human relevance, positively (+) or negatively (−) ranking a candidate gene (Table S1).

## sgRNA and crRNAs target site selection

*rrad* (ENSORLG00000024517) sgRNAs, *adprhl1* (ENSORLG00000004693) sgRNAs, *ptprd* (ENSORLG00000004685) sgRNAs, *phka2* (ENSORLG00000003555) sgRNAs, *blzf1* (ENSORLG00000003670) sgRNAs, *btbd1* (ENSORLG00000003434) sgRNAs, *zrsr2* (ENSORLG00000003476) sgRNA, *pcdh17* (ENSORLG00000004535) sgRNA, *sec61a1* (ENSORLG00000016830) sgRNA, *irf1a*(ENSORLG00000011716) sgRNA, *gse1* (ENSORLG00000009542) sgRNA, and *gpc5a* (ENSORLG00000026952) sgRNA were designed with CCTop and ACEofBASEs as described previously[44,52], on the medaka genome in Ensembl release 101 (Japanese medaka HdrR assembly, Aug 2020), Ensembl release 102 (Japanese medaka HdrR assembly, Nov 2020), Ensembl release 103 (Japanese medaka HdrR assembly, Feb 2021) and Ensembl release 106 (Japanese medaka HdrR assembly, Apr 2022), respectively. The target sites and oligonucleotides selected for sgRNA cloning are listed in Tables S4 and S5. Cloning of sgRNA templates and in vitro transcription were performed as described previously[52]. The plasmid DR274 used was a gift from Keith Joung (Addgene 42250). For base editing, locus-specific crRNAs and the tracrRNA backbone were obtained from IDT (custom Alt-R crRNA). crRNA (100 μM) and tracrRNA (100 μM) were diluted in nuclease-free duplex buffer (IDT) to a final concentration of 40 μM and incubated at 95 °C for 5 min.

## In vitro transcription of mRNA

The plasmid pCS2 + (Cas9) and pCS2 + (evoBE4max) were linearized using NotI, and mRNA in vitro transcription was performed using the mMESSAGE mMACHINE SP6 or T7 Transcription Kit (Thermo Fisher Scientific) and purified with the RNeasy Mini Kit (Qiagen).

## Microinjections

Microinjections were performed in HdrR (*myl7::EGFP*, *myl7::H2A-mCherry*) zygotes. The Cas9 injection solution contained 150 ng/μl Cas9 mRNA, 10 ng/μl sgRNA, and 10 ng/μl GFP mRNA as injection tracer. The base editor injection solution contained 150 ng/μl evoBE4max mRNA, 4 pmol sgRNA, and 10 ng/μl GFP mRNA as injection tracer. Control injections were performed with 10 ng/μl GFP mRNA as injection tracer. Injected embryos were maintained at 28 °C in medaka embryo rearing medium (ERM, 17 mM NaCl, 40 mM KCl, 0.27 mM CaCl2, 0.66 mM MgSO4, 17 mM Hepes) and selected for GFP expression 7 h post-injection. Phenotypes were assessed 4 days post-fertilization. For sarcomere staining, microinjections were performed in Cab embryos with similar Cas9 conditions as described above.

## Image acquisition and phenotyping of knockout models

Embryo morphology and heart dynamics were documented with a Nikon SMZ18 equipped with Nikon DS-Ri1 and DS-Fi2 cameras. Embryos were mounted in EMR into injection molds (1.5% (w/v) agarose in ERM). Heart rate quantification of CRISPR-Cas9 and evoBE4max-mediated knockout embryos and control embryos was obtained using an ACQUIFER Imaging Machine and the HeartBeat software as described previously[20,53]. To acquire the heart rate, fluorescent image sequences have been captured for 10 s with 24 fps at 21 °C, 28 °C, and 35 °C.

## Immunostaining of knockout models for cardiac sarcomere structures

After microinjection, embryos were screened for cardiac phenotypes at stage 32 (4 dpf). Overnight fixation was done in 4% paraformaldehyde in PBST (0.1% Tween in phosphate-buffered saline (PBS)). Immunostaining of sarcomere structures was performed using myl7 primary antibody (Genetex, Catalog no GTX128346-200) in 1:500 dilution in PBST. Fluorescently labeled secondary antibody anti-rabbit 488 (Life Technologies, Catalog no A-11034) was used in 1:500 dilution in PBST. Embryos were imaged using Leica Sp8 confocal microscope, and images were processed using Fiji[54].

## Genotyping

Single phenotypic, non-phenotypic, and control embryos were lysed in DNA extraction buffer (0.4 M Tris/HCl pH 8.0, 0.15 M NaCl, 0.1% SDS, 5 mM EDTA pH 8.0; 1 mg/ml proteinase K) at 60 °C overnight. Samples were diluted 1:2 with nuclease-free water, and the proteinase K was heat-inactivated afterwards for 20 min. Precipitation of genomic DNA was done in 300 mM sodium acetate and 3x vol. absolute ethanol at $20,000 \times g$ at 4 °C, followed by resuspension in TE buffer (10 mM Tris pH 8.0, 1 mM EDTA in RNAse-free water). The target loci were PCR amplified in 30 PCR cycles using Q5 High-Fidelity DNA Polymerase (New England Biolabs), locus-specific primer pairs (Table S6), and 1 μl precipitated DNA sample. PCR products were purified after agarose gel electrophoresis using the Monarch DNA Gel Extraction Kit (New England Biolabs) and submitted for Sanger sequencing to Eurofins Genomics. Base editing results were analyzed with EditR[55].

## Vibratome sectioning of adult medaka strains

HO5 and HdrR adult medaka were euthanized in 20% tricaine for 2 h at room temperature and then fixed in 4% paraformaldehyde in PBST (0.1% Tween in phosphate buffered saline (PBS)) overnight at 4 °C. Samples were embedded in 45% albumin, 1.5% gelatin and 25% glutaraldehyde solution in 12:6:1 ratio and sectioned with 70 μm thickness using a Leica VT1000S vibratome. Staining of sarcomere structures was performed on the sections with 4′,6-Diamidin-2-phenylindole (DAPI) in 1:500 dilution in PBST. Sections were mounted in 60% glycerol in PBS and imaged using Leica Sp8 confocal microscope.

## Data analysis and statistics

Data visualization and statistical analysis were performed using R[56].

## Reporting summary

Further information on research design is available in the Nature Portfolio Reporting Summary linked to this article.

## Data availability

All data supporting the findings of this study are available within the paper and its Supplementary Information. Source data are provided with this paper (Data S1–S4). Raw data are also available upon request to the corresponding authors. The individual raw sequencing datasets are linked to the following project IDs: Illumina DNA sequencing data (PRJEB17699). https://www.ebi.ac.uk/ena/browser/view/PRJEB17699. Illumina RNA sequencing data (PRJEB43091). https://www.ebi.ac.uk/ena/browser/view/PRJEB43091.

## Code availability

The full code is available at the indicated GitHub repository (https://github.com/tf2/ViteRbi) and additional information is available upon request to the corresponding authors.

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

## Acknowledgements

We thank Steffen Lemke and Lazaro Centanin for critically reviewing the manuscript and the Wittbrodt, Birney, Lemke, and Centanin labs for constructive support of the work and manuscript. Thanks to R. Hodge for critically reading and commenting on the manuscript. We thank J. Gehrig (ACQUIFER, Bruker) for expert technical advice in high-throughput imaging and benchmarking. We thank T. Kellner, B. Wittbrodt, and R. Müller for excellent technical support, as well as E. Leist, M. Majewski, A. Saraceno, and S. Erny for fish husbandry. Funding sources: European Research Council (ERC) Synergy grant 810172/IndiGene (EB, JW). National Institutes of Health (NIH) grant R01ES029917 (EB, JW). German Centre for Cardiovascular Research (DZHK) grant 81X2500189 (JG, JW, NH). Deutsche Herzstiftung e.V. grant S/02/17 (JG). Research Center for Molecular Medicine (HRCMM), Career Development Fellowship (JG). Medical Faculty Heidelberg University, MD/PhD program (JG). Joachim Herz Stiftung, Add-On Fellowship for Interdisciplinary Science (JG).

## Author contributions

Conceptualization: J.G., K.N., E.B., and J.W.; Investigation: J.G., B.W., T.F., T.T., R.A., O.H., A.L., P.W., K.N., D.H., N.H., E.B., and J.W.; Visualization: J.G., B.W., T.F., T.T., and A.L.; Funding acquisition: J.G., N.H., E.B., and J.W.; Project administration: D.H., E.B., and J.W.; Supervision: D.H., N.H., E.B., J.W.; Writing—original draft: J.G., B.W., T.T., T.F., E.B., and J.W.

## Funding

## Competing interests

The authors declare no competing interests.
