## [Transparent Peer Review file · Nature Communications]

Natural genetic variation quantitatively regulates heart rate and dimension

Corresponding Author: Professor Joachim Wittbrodt

Version 0:

Reviewer comments:

Reviewer #1

(Remarks to the Author)

Review of Nature Communications manuscript Gierten et al, March 2204

This is a very interesting study, on a highly relevant topic. I do think a more thorough functional assessment of the mutants in this study is needed, to support some of the claims made.

In general, the paper is well-written and the figures are well put together. However, some sections are not so well written, which makes it a hard read at times.

Gierten et al describe the use of inbred medaka strains to detect new genomic loci associated with heart rate variability. Using 5 different isogenic Medaka strains they settle on two strains with the largest difference in heart rate at different time point during cardiac development.

Making use of a simple heart rate assay they analyse the heart rate in the 5 different strains, and the authors state that they using strains from two geographically very distinct regions in Japan, they gain the highest genomic variance. Surprisingly they find two strains, HdrR and HO5, both from the northern part of Japan, to show the largest difference in heart rate.

Incrossing the two selected strains, the authors make use of genomic analyses to make a SNP map of the genetic variance present in these two lines.

Follow up experiments with genome-wide QTL mapping and gene enrichment analyses, they narrow down the number of loci to 59 and region on chromosome three that is highly associated with heart rate variability and development.

Fine mapping and RNA sequencing is used to further narrow down the genomic loci to 12 genes of interest, some of which have not been described in the context of cardiac development and function.

I have the following comments and questions, that I hope the authors will address during the revision process:

the authors start by describing the HLHS and the importance of elucidating the genetics behind this, genetically, poorly understood disease. However, most of the paper is focused on heart rate. Very little is done to look at the developmental cardiac phenotypes present in the KO models. The authors seem to draw a direct parallel between heart rate differences and cardiac development.

Do the authors claim that there is no need to separate the two phenotypes? Do any of the KOs with heart rate variability show normal cardiac development? And does any of the KO with defects in cardiac development show normal heart rates?

The HO5 line is described with an unbalanced heart. What does the authors mean by this statement? Unbalanced in rhythm or development?

In the swim tunnel, the HO5 line perform significantly worse than the HdrR strain. The authors describe the body movement used to generate swimming movements as being perturbed in the HO5 strain, and conclude that this is due to poor cardiac

function.

Have the authors done any analysis of skeletal muscle structure in these fish? The poor performance in swim tunnel could be due to skeletal muscle defects rather than cardiac defects.

In line 338 the authors write "heartbeat detection surfaced cardiac conduction". What is meant by this?

The simple cardiac analysis that has been done in the study is not enough to claim that the KOs show electrophysiological phenotypes. The AV-block does indeed suggest that there is a conduction defect between the atria and ventricle, but further analyses is needed for the authors to make a claim to the rrad crispants re-capitulate that of a Brugada syndrome in vivo model.

The genes identified have in some cases been associated with sarcomere structure and development. Does the mutants show any sarcomere defects?

Reviewer #2

(Remarks to the Author)

General comments

This is an interesting manuscript using an intercross between two strains of Medaka to generate a genetic map of heart rate QTLs. Notably, similar experiments have been undertaken for multiple other species, and it is not easy to see from the current manuscript where the true novelty lies. The title is quite declarative yet seems to ignore the prior decades of work which the current manuscript confirms. Indeed, the full utility of scale in a vertebrate that the medaka might bring to QTL analysis has not been fully utilized (see detailed comments). The experiments are well conducted and the results are clearly presented, but their incremental value is diluted by some of the assumptions made, the specific experimental design choices (including the absence of several critical downstream experiments) and the rather imprecise nature of the introduction and discussion. The manuscript confounds a series of physiologic terms which highlight the limitations of the experimental design and diminish the manuscript's implications for the traits outlined or for QTL analyses in general.

Specific comments

The introduction elides some of the core issues in quantitative trait definition and analysis. Indeed, cardiac physiologic and structural phenotypes, which are among the most common congenital abnormalities, are among the most useful exemplars of the dependence of genetic analyses on the resolution of phenotypic measurement. Traits which can appear continuous at the level of a population often become discontinuous as genetic mapping is undertaken (this has largely limited the ability to reduce QTLs to mechanistic variants in mouse or other species). Indeed, the examples where QTLs have led to important mechanistic insights have occurred where a specific QTL can readily be decomposed on more rigorous analysis into constituent binary traits, usually when formal segregation analyses are feasible at the relevant scale (PMID: 11907579). Thus, while ventricular size may appear as a QTL at a population level, it is clinically manifest as traits like HLHS a range of mitral, ventricular, outflow tract and aortic abnormalities with autosomal dominant transmission and reasonable penetrance but variable expressivity which is usually not detected except in formal proband-driven familial phenotyping (PMID: 29106500). Indeed, this fundamentally superficial phenotyping is likely one contributor to the paucity of causal genes identified in congenital heart disease despite demonstrably high levels of familial recurrence for several of the most important syndromes (PMID: 29106500). Given the medaka trait they outline appears an interesting potentially tractable model for HLHS (including reduced survival to adulthood), it might have been more informative to have deeply phenotyped this strain and then mapped the genetic and environmental modifiers of survival in subsequent crosses.

The introduction might usefully describe the limitations and confounding observed with quantitative traits and their subsequent mapping in other organisms. The complexity of true quantitative traits (as observed in the current models) is such that individual QTLs often break down in subsequent analyses where multiple genetic contributions within a single locus cannot be further reduced as the phenotype (quantitative trait variation) dissipates with any recombination within the locus haplotype. This has led to systems level analyses in other organisms for heart rate and other such traits. As noted above, when QTL approaches can be resolved through massive segregation studies for higher resolution phenotypes, there is the potential to resolve QTLs at a mechanistic level.

The section entitled heart phenotype contrasts also seems to elide different attributes of heart rate and cardiac function which are already extremely well understood. For example, there is no discussion of the effects of extracardiac physiology (e.g. neuronal, skeletal muscular or metabolic) on heart rate, even during early development. Cardiac structural morphology is a broad descriptor for a wide range of abnormalities with discrete effects on heart rate and cardiac function presumably refers to contractility, relaxation etc, which also have some competing influences on heart rate through very discrete mechanisms and which might have been discerned in parallel during assessment of the parental strains and would have assisted in choosing the optimal experimental design to explore heart rate as a 'pure' quantitative trait.

It is not clear that plate address was controlled for in the data acquisition and analysis but even in a regulated chamber the light necessary for video acquisition has led to temperature differences which over a 96 wellplate can be substantial.

The authors note discrete swimming performance between strains but fail to discriminate between cardiac and skeletal muscle contributions to this phenotype. Is the heart rate phenotype in the HO5 strain purely cardiac or is there also a skeletal muscle abnormality. The failure to account for orthogonal phenotypes renders subsequent heart rate analyses difficult to

interpret.

Segregation analysis

The authors perform an F1 intercross and then sample '1260 F2 individuals unique inherited genotypes'. It is not clear what this sentence means. Were genotypes used to dictate the sampling? Were haploid fish generated and then offspring sampled or is this just 1260 F2 individuals all of whom as in any F2 would be assumed to have "unique inherited genotypes" on the basis of normal meiosis?

Genome-wide QTL mapping

The authors then undertake a standard QTL analysis using the recombinant panel they have created among the 1260 F2 offspring. The rationale for now exploring temperature sensitivity is unclear when these data are not presented for the parental strains.

There is a clear over representation of H05 alleles which is somewhat concerning given the discrete gross phenotypes and the discrete viability of this strain as described. This is attributed to the training set used for the Markov Model, but the potential plausible biological causes could also have been formally tested with a single additional cross.

The issues with QTL mapping are highlighted nicely in the analyses they outline. Without discrete recombinant mapping to much higher resolution, one has difficulty in identifying the transmissibility of the trait components associated with individual haplotype blocks and the variants or genes involved cannot predictably be defined. Indeed, all of the issues with association mapping are amplified here because both trait imprecision and locus imprecision compound. Trait segregation patterns cannot be rigorously defined and no part of the locus can be discretely associated with the trait (it could readily be an intergenic regulator of genes far outside of the block associated with the trait). To overcome these challenges one could continue to scale the generation and analysis of informative meiotic events, but as is typical in such studies the jump is made to use less rigorous (and less costly and time-demanding) approaches including gene expression analyses or testing of generic LoF alleles both of which make quite substantial assumptions about mechanism which are never then tested. Examples from yeast are particularly informative in this context where single 'QTLs' may be resolved into discrete gain of function alleles only a few base pairs apart with very large effect sizes that are simply undetectable without rigorous segregation to variant level (PMID: 11907579). The potential to execute on this in a vertebrate is at present largely restricted to fish species and might have been usefully undertaken with one additional cross.

The gene enrichment analyses are perfectly well executed and described. Their relevance to the remainder of the manuscript is largely by word association rather than experimental proof. The authors assume that the genes responsible for their heart rate observations are differentially expressed in a particular time window in a single tissue and are physically located within recombination boundaries that are statistically defined.

In vivo validation

The authors test CRISPR LoF in a series of candidate genes defined by the locus boundaries and the gene enrichment analyses. There is no formal demonstration of the mode of transmission or of any other analyses which might support such a model. How were gain of function effects excluded?

Many of these issues (which affect all such QTL studies) could have been at least partially addressed by closing the phenotypic loop for the heart rate traits. For example if there are heart rate phenotypes with a specific gene LoF which are mediated via discrete effects on variable AV delay this could then be studied in the relevant parental strains to see if discrete mechanisms are transmitted at a phenotypic level even if not formally shown at a genetic level. Allusions are made to different mechanisms for human traits associated with some of the genes but none of these mechanisms (automaticity differences, conduction block, repolarization prolongation, etc) are then tested in the medaka model.

The authors might consider how likely the average gene not selected by the process they describe would be to have a generic effect on heart rate. Such a control group could have been tested for each step (genetics, gene enrichment and in vivo testing) for generic heart rate or for high resolution mechanistic phenotypes. The current manuscript does not include any such controls.

Discussion

The discussion is perfectly reasonable but glosses over the methodologic challenges noted above. These challenges are undoubtedly not unique to this study but the core message of the manuscript is that somehow access to fish radically changes the rigor of the genetic dissection of human disease traits by "partitioning and validating the spectrum of phenotypes in a strain specific way". The manuscript highlights the confusion that appears to exist in modern genetics between genotype and phenotypic architecture. As noted in the comments on the in vivo validation section, one thing that this manuscript does not even attempt is to shed any light on the partitioning of phenotypes. Among the missed opportunities in the current study is the chance to resolve discrete strain-specific effects on a high-level trait like heart rate into discrete phenotypes and then to map these to individual variants by which mechanism might be discerned.

Version 1:

Reviewer comments:

Reviewer #1

(Remarks to the Author)

The authors have addressed my comments with regards to the written part of the manuscript. They have also addressed most of my concerns and questions regarding the missing experimental data.

However, I am a bit uncertain as to why the mutant line carrying a mutation in RRAD, the most promising gene from a human disease point of view, has not been characterized farther?

Since this gene has previously been associated with severe myofibril disarray (PMID: 33195237) in cultured cardiomyocytes, as well as calcium mishandling, it is surprising that the authors have not included the mutants in the sarcomere analyses done on the other 4 mutants.

The myofibril disarray would be a simple add-on to show that the in vivo knock down recapitulate the phenotype observed in cell culture, and add to the significance of the screening approach the authors present in the current study.

(Remarks on code availability)

Reviewer #2

(Remarks to the Author)

The authors have responded to each of the reviewers' comments, and while they have undertaken some additional experiments, the bulk of their responses are simply restatements of their original output with some additional results and no new supportive data. The data presented simply do not support the conclusions they have drawn. Specific comments are outlined below.

The lack of novelty is addressed by reframing their initial claims so narrowly so as to reinforce the original comment. In essence they note that this is the first study with two naturally originating isogenic Medaka lines (as for all prior mouse and fly studies). They also somehow suggest that 'naturally originating' isogenic lines are more representative of outbred human data. Obviously, the prior studies with isogenic mouse and fly genetic studies have as much relevance to observations in outbred humans as the current dataset since they also are originally derived from naturally occurring alleles. They also confound quantitative traits in human GWAS (where there are loci associated with contributions to such traits through linkage disequilibrium but no evidence of transmission) with QTLs in an inbred cross (where the transmission and segregation enables locus definition using recombinants). Finally, the authors suggest they have produced new in vivo experiments, but it is not clear to this reviewer to which experiments they refer.

There is a pervasive failure in the manuscript to distinguish between genomics and genetics. For example, it is simply not possible to define the mode of inheritance from single generation genomics. The PMID citation noted in my original comment clearly demonstrates the inheritance mode of HLHS (among other congenital heart disease traits) is autosomal dominant from familial aggregation studies. Indeed, I cited this study precisely because it highlights how lack of rigor in phenotyping (ie assuming that every carrier of the causal genotype in the kindred will have exactly the same phenotype as the proband) underlies the inability to map and clone these conditions in humans (as in medaka and any other species). Their interpretation of these findings highlights the lack of discrimination between familial aggregation and transmission likelihood on one hand and measured variation across the genome on the other. With the degree of familiarity exhibited the mode of inheritance cannot be polygenic under normal transmission probabilities.

I agree that the potential for Medaka to resolve true QTLs (if they can be resolved at all) is relevant, my point is simply that the current manuscript does not resolve these to any greater extent than prior mouse or fly data despite the potential for definitive resolution through exactly the deep phenotyping which I suggested in my review. To be more precise, one hypothesis for the difficulties in resolving QTLs is the focus on a single quantity. In other words if one were to explore heart rate in conjunction with other orthogonal traits such as ventricular mass, AV conduction or Ca²⁺ handling, then one might be able to define robust genetic support (as opposed to genomic association) for specific variants in specific genes and then model how these various loci interact to generate a final output (see also below on Brugada claims).

Genome editing is just that. Individual alleles may have gain or loss of function or both for the gene of interest. Knocking out a gene with CRISPR is helpful validation if you have data that the trait which you are trying to model has the relevant inheritance characteristics which suggest dose dependent loss of function, but many individual mutations even in large effect size Mendelian disorders have quite clear effects through loss of some functions and gain of others. Otherwise, gene editing simply shows that the gene is involved in some way in cardiac biology-in particular if the phenotype is so superficial as simply any effect on heart rate. Even the sodium channel gene which the authors refer to in their discussion of Brugada syndrome exhibits variation which has effects on multiple different functions (conductance, coupling, excitability, force transduction) in different disease syndromes.

The response which invokes physiologic layers further highlights the problems outlined above. The authors assume that there is a discrete relationship between heart rate and the other layers they cite, but my point is that there are multiple different ways to generate any particular effect on heart rate and without looking at the other layers, one is forced to assume that each heart rate has some distinct validity which is already known not to be the case (see multiple ion channels, calcium handling protein and junctional molecules which result in slowing of heart rate in human disease yet can all be discriminated by associated orthogonal phenotypes). The authors appear to recognize these limitations but suggest that they are

overcome by “direct state of the art gene editing techniques” when they can only be resolved by extensive additional phenotyping.

The appropriate statistical approach for assessing ‘within plate’ variance in screening is a formal z-score.

Among the new data in the revised manuscript appear to be skeletal muscle histology in the H05 adults. Obviously, this does not address functional performance in any direct way, but also highlights the generally superficial nature of the phenotyping rationale across the manuscript. Low cardiac output is also observed but the presence of this one phenotype does not preclude any or many others.

The authors do clarify their intercross and genotyping strategy while also noting prior data on temperature sensitivity in the parental strains.

The authors agree there is overrepresentation of the H05 allele but somehow feel this is not the focus of a study whose fundamental conclusions are based on allele transmission with heart rate. This might be a central finding of the study-if there is some gene-gene interaction which favors H05 (possibly a survivorship bias or selection of HLHS prone alleles due to some developmental constraint) it might be the key to the presence of such potentially lethal alleles in the human population). The lack of follow up of this genotype imbalance substantially weakens any conclusions.

The authors note the challenges with their transcriptomic data but have included these data without validation.

Surely the generic modeling of arbitrary alleles rather than the specific mechanistic effect of the naturally occurring alleles confounds the premise by which the authors claim novelty for their model and specific experiments. It should be noted that the generation of isogenic lines already selects for very specific subsets of mechanism.

The section on RRAD further highlights the laxity in phenotyping and terminology which characterizes the current manuscript. The authors note that regional conduction abnormalities are seen in the ventricles of Brugada subjects but then suggest that AV conduction (a completely distinctive developmental tissue not shown to be involved in Brugada related AV block which is where reported infraHisian and probably junction related rather than channel conductance related, though there are other sodium channel disorders which are associated with AV block-once again all confounded by low resolution phenotyping in both human and in this instance Medaka) represents a surrogate for this highly specific disease phenotype. This could be true, but AV block per se does not contribute to ventricular arrhythmogenesis in Brugada syndrome, and the authors also assume that the AV block observed is orthologous without any new data.

There is also an effort to characterize the effects of editing 3 genes on cardiomyocyte structure as highlighted by some of the data presented for *adphr1*. It is not obvious how this assists the message of the manuscript since this is but one such potential phenotype among many which might have been tested.

The authors note that the control data were not collected in parallel but rather date from a prior set of experiments. This essentially renders much of the work presented difficult to interpret given the effects of selection and the wide range of mechanisms by which heart rate might be affected.

The final allusion in the authors’ response to reviews is that they are simply setting the stage for subsequent work. As noted, the results of these downstream experiments would directly address the uncertainty currently clouding their conclusions which remain incompletely substantiated by the data presented.

(Remarks on code availability)

There are literally tens of such algorithms in the literature (even just for fish)-the majority based on the same basic mathematical transform.

Version 2:

Reviewer comments:

Reviewer #3

(Remarks to the Author)

(Remarks on code availability)

Point by point response to the reviewers' comments:

We thank both reviewers for their reports and valuable comments, which have contributed substantially to enhance the quality of our manuscript. Below we give a point-by-point response to all issues raised and how we have addressed them in the revised version of our manuscript. To facilitate the review of the revised manuscript, we have underlined any significant changes or additions to the text in the manuscript and supplemental information. We are confident that these improvements fully address all reviewers' comments and suggestions.

Reviewer #1 (Remarks to the Author):

This is a very interesting study, on a highly relevant topic. I do think a more thorough functional assessment of the mutants in this study is needed, to support some of the claims made.

In general, the paper is well-written and the figures are well put together. However, some sections are not so well written, which makes it a hard read at times.

Gierten et al describe the use of inbred medaka strains to detect new genomic loci associated with heart rate variability. Using 5 different isogenic Medaka strains they settle on two strains with the largest difference in heart rate at different time point during cardiac development.

Making use of a simple heart rate assay they analyse the heart rate in the 5 different strains, and the authors state that they using strains from two geographically very distinct regions in Japan, they gain the highest genomic variance. Surprisingly they find two strains, HdrR and HO5, both from the northern part of Japan, to show the largest difference in heart rate.

Incrossing the two selected strains, the authors make use of genomic analyses to make a SNP map of the genetic variance present in these two lines.

Follow up experiments with genome-wide QTL mapping and gene enrichment analyses, they narrow down the number of loci to 59 and region on chromosome three that is highly associated with heart rate variability and development.

Fine mapping and RNA sequencing is use to farther narrow down the genomic loci to 12 genes of interest, some of which have not been described in the context of cardiac development and function.

I have the following comments and questions, that I hope the authors will address during the revision process:

We thank the reviewer for this very positive and constructive feedback. We appreciate the raised questions and give a point-by-point response to the specific comments below and how we have addressed them in the revised version of our manuscript.

the authors start by describing the HLHS and the importance of elucidating the genetics behind this, genetically, poorly understood disease. However, most of the paper is focused

on heart rate. Very little is done to look at the developmental cardiac phenotypes present in the KO models. The authors seem to draw a direct parallel between heart rate differences and cardiac development.

Do the authors claim that they're is no need to separate the two phenotypes?

We appreciate this consideration and thank the reviewer for the opportunity to clarify this conceptually highly relevant point. Cardiac (morphological) developmental phenotypes and heart rate are separate entities yet are interrelated. We introduced HLHS as a congenital phenotype of prominent interest given its severity and persisting significant challenges in the care of affected children and specifically as it represents a prime example of a quantitative phenotype with trait distributions reflecting a gradual transition from “still healthy” to pathological. The thought-provoking point is that nearly any cardiac phenotype in congenital heart disease can be described more accurately using quantitative metrics. Consequently, there is no clear-cut boundary between disease and health. This is why we indicated a couple of “physiological traits”, likewise underlying quantitative variation. These quantitative phenotypes are nowadays perceived to be widely related to polygenic factors that are interrelated with environmental triggers. Generating a better understanding of the underlying genetics of such quantitative trait variation is as challenging as it is highly relevant to individualized medicine. Interestingly, genetic variations mapped in the “healthy spectrum” apply to the “diseased spectrum” and vice versa.

Thus, to tackle polygenic contributions and identify specific variants and genes associated with physiological and diseased phenotypes, we intentionally focused on the heart rate phenotype, given its central role and significance and precisely scorable output reflecting interactions of different types of cardiac properties, “including electrophysiology, morphology, and function.”

Of note, profiling embryonic heart rates in different medaka inbred strains led us to identify not only differences in heart rate levels but also significant quantitative morphological differences, i.e., hypoplastic ventricle in the HO5 strain, demonstrating the robustness of the heart rate parameter as a proxy to reflect multiple phenotypic layers with developmental manifestation. Consequently, mapping heart rate metrics implies genes associated with genes that impact cardiac development through different mechanisms, the result and relevance of which we tested by direct state-of-the-art gene editing techniques.

Do any of the KOs with heart rate variability show normal cardiac development?

All Cas9 or base editor specimens in Figure 4, screened for successful microinjection, were embryos showing normally developed hearts. We intentionally focused on this normal heart development population to exclude secondary effects in morphologically affected embryos. In the revised version of the manuscript, we provide additional data and have included an additional supplemental figure now also presenting the heart rates of the phenotypically affected morphants.

And does any of the KO with defects in cardiac development show normal heart rates?

We thank the reviewer for addressing this point - to clarify it, we provide new experimental data in the new Figure S6, showing Cas9- and base editor-mediated gene editing experiments for all test genes with automated heart rate readout specifically on the subpopulation with morphologically affected heart development. This additional data set demonstrates a higher number of gene knockouts with significant effects on heart rate metrics over the three test temperatures (for 12 candidate genes, 54 conditions were tested across three different temperatures and gene editing tools. Crispants and editants with impaired cardiac

development showed altered heart rates (highlighted in red) in 26 conditions tested, while those with normal cardiac development exhibited a heart rate effect in 19 conditions tested); this finding underlines our approach to analysis heart rate effects of test genes in embryos with morphologically unaltered heart development, cf. main manuscript - added and modified text: *“To assess the primary effects of edited genes on heart rate, we profiled heart rates in successfully injected (Cas9 and base editor) embryos and measured heart rates at three different temperatures. In the two groups with apparently normal heart development and cardiac phenotypes (cf. Fig. 3), we found a higher number of genes showing significantly altered baseline heart rates over the test temperatures and across gene editing methods in cardiac affected embryos, likely reflecting secondary effects of aberrant morphology and function on heart rate (Fig. S6).”* and *“Thus, to exclude secondary effects in embryos with aberrant cardiac morphogenesis, we applied the automated heart rate assay in embryos with apparently normal heart development (Fig. 4, Fig. S6).”*

The HO5 line is described with an unbalanced heart. What does the authors mean by this statement? Unbalanced in rhythm or development?

Here we referred to the development. We thank the reviewer for making us aware of this unclear terminology; “unbalanced” stems from the medical domain, specifically from a certain type of (human) congenital heart disease, in which one of the ventricles is hypoplastic and was used to describe the hypoplastic ventricle phenotype in HO5. To avoid confusion, we modified the sentence *“HO5 embryos exhibited unbalanced heart chambers with a dominant atrium and an underdeveloped (hypoplastic) ventricle”* to *“HO5 embryos exhibited disproportional heart chambers with an enlarged atrium and underdeveloped (hypoplastic) ventricle”*.

In the swim tunnel, the HO5 line perform significantly worse then the HdrR strain. The authors describe the body movement used to generate swimming movements as being perturbed in the HO5 strain, and conclude that this is due to poor cardiac function.

We thank the reviewer for this valuable comment, prompting us to specify wording in this paragraph to clarify the apparent confusion. Given the specific embryonic cardiac phenotype in the isogenic HO5 strain, we meant to assess the physical fitness of this strain and the cardiac function at an adult stage in this fixed genotype. We now write, *“As the unique combination of natural genetic variation fixed in the HO5 genome contributes to its cardiac phenotype, we next assessed the extent to which the genomic composition influences the physical fitness and cardiac function at adult stages.”*

Have the authors done any analysis of skeletal muscle structure in these fish?

The poor performance in swim tunnel could be due to skeletal muscle defects rather than cardiac defects.

Taking advantage of this valuable comment, we generated and analyzed skeletal muscle vibratome sections. Microscopic analysis of skeletal muscle sections revealed histologically intact skeletomuscular organization in adult HO5 fish, thus not providing evidence for a primary skeletal muscle phenotype underlying the poor swimming performance. However, the constitution of adult HO5 (relatively short and round fish) might also play into the movement pattern and performance. At the same time, our echocardiography readout demonstrates low cardiac output, which is likely directly related to physical fitness. We modified this section in the revised manuscript to put these observations into the context of additional histological analysis, provided in new Figure S1.

In line 338 the authors write “heartbeat detection surfaced cardiac conduction”. What is meant by this?

“Surfaced” in the sense of revealed - sentence was adjusted to “revealed”.

The simple cardiac analysis that has been done in the study is not enough to claim that the KOs show electrophysiological phenotypes. The AV-block does indeed suggest that there is a conduction defect between the atria and ventricle, but further analyses is needed for the authors to make a claim to the *rrad* crispants re-capitulate that of a Brugada syndrome in vivo model.

We appreciate this comment, which helped to specify our considerations. Interestingly, the genetic disorder Brugada syndrome, although classically associated with mutations in the sodium channel gene *SCN5A*, remains genetically ill-defined as many gene associations indicate complex genetics in familial cases. While different efforts were started to define the causality for some of these genes, often with ion channel-regulating function, numerous new potential genetic players are not yet assigned to specific Brugada syndrome subtypes. We were excited to find the *rrad* gene tightly associated and to uncover a specific conduction phenotype in gene editing experiments. Our validation showed a clear genotype-phenotype correlation underpinning *rrad* as responsible player impacting on conduction. This *in vivo* finding (stable over generations) is particularly relevant given the association of the human ortholog *RRAD* as a new susceptibility gene detected in familial Brugada syndrome cases (Belbachir et al. 2019, PMID: 31114854), with cardiomyocytes derived from induced pluripotent stem cells expressing pathogenic *RRAD* variant exhibiting electrophysiological features of Brugada syndrome and cytoskeleton disturbances. For obvious reasons, the arrhythmic manifestation of Brugada-associated gene mutations in a two-chambered fish heart and a four-chambered human heart might differ considerably in terms of the anatomical location of the arrhythmia. As such, the exact recapitulation of the electrophysiological phenotype is not the primary motivation for studying gene function in the medaka model but rather for testing the general mechanistic role and *in vivo* relevance of a gene of interest.

Moreover, while Brugada syndrome is known for its propensity to ventricular arrhythmia, slow electrical conduction in subareas of the heart is thought to represent a major electrical pathological substrate. In this regard, it is a particularly relevant finding that *rrad* knockout models in medaka showed atrioventricular blocks, likely as a manifestation of a slow conduction phenotype. Additional significance to this finding is added by observations that Brugada patients can develop severe (high-risk) atrioventricular blocks (AVB); intrinsic conduction abnormalities were discussed as one of the potential mechanisms for AVB, while

the genetics in Brugada patients at risk remain elusive (Kamakura et al. 2021, PMID: 33428312). We provide new *in vivo* evidence for *rrad* mutations leading to AV block, which could have immediate prognostic and therapeutic value for a subset of Brugada patients, as Brugada patients with high-risk AVB should be managed by cardiac pacing (which has implications for the choice among different types of implantable cardioverter defibrillators). We thank the reviewer for this input, which enabled us to restructure the related section in the discussion to re-focus on our relevant findings.

The genes identified have in some cases been associated with sarcomere structure and development. Does the mutants show any sarcomere defects?

Acknowledging this essential question, we performed additional experimental work. We established a new series of crispants and analyzed the sarcomeric structure in the respective hearts. Based on the decreased ventricular size in crispants and cardiac expression (own RNA seq data), we have evaluated the sarcomeric structure in *adprh1*, *blzf1*, and *btbd1* crispants. Targeted gene editing (Cas9) and cardiac immunostaining with myl7 (regulatory myosin light chain) antibodies at the relevant stage 32 (4 dpf) showed disorganized sarcomeric patterns in *adprh1* crispants (new Figure 5C), further supporting its role in myofibrillar assembly initially suggested by its localization to the cardiac sarcomere in *Xenopus* (Smith et al. 2016, PMID: 27217161).

Reviewer #2 (Remarks to the Author):

General comments

This is an interesting manuscript using an intercross between two strains of Medaka to generate a genetic map of heart rate QTLs. Notably, similar experiments have been undertaken for multiple other species, and it is not easy to see from the current manuscript where the true novelty lies. The title is quite declarative yet seems to ignore the prior decades of work which the current manuscript confirms. Indeed, the full utility of scale in a vertebrate that the medaka might bring to QTL analysis has not been fully utilized (see detailed comments). The experiments are well conducted and the results are clearly presented, but their incremental value is diluted by some of the assumptions made, the specific experimental design choices (including the absence of several critical downstream experiments) and the rather imprecise nature of the introduction and discussion. The manuscript confounds a series of physiologic terms which highlight the limitations of the experimental design and diminish the manuscript's implications for the traits outlined or for QTL analyses in general.

We thank the reviewer for the encouraging assessment of our work and the constructive feedback. We also appreciate the critical comments, which have made us realize that some aspects of our work needed to be presented clearly and discussed more in the original version of our manuscript.

Our approach to resolving heart rate QTLs in a medaka inbred genetics resource is novel and unique. While medaka is genetically very appealing and offers several properties that allow strict phenotyping, it had yet to be used to study human-relevant cardiac phenotypes at a large scale, specifically quantitative cardiac traits. With this study, we introduce this new approach and demonstrate its success in new genetic findings not captured or validated by previous complementary studies in human or other model systems.

We absolutely agree that there have been numerous impactful studies over many years in linkage or association mapping, all of which have advantages and obstacles. Human GWAS has been very successful in discovering many loci in complex traits, and we cited intentionally a selection of substantial GWAS on heart rate (cf. den Hoos et al. 2013, PMID: 23583979; Epinga et al. 2016, PMID: 27798624), as we also use heart rate as a central phenotype. However, it is common sense that significant challenges exist in outbred human genetics, including the straight forward experimental validation and „*the variability in individual phenotypes and genotypes, environmental factors, genetic relatedness and population stratification*“, as indicated at the end of the first introduction paragraph; specifically, uncontrollable GxG, GxE and potential GxGxE in humans may act as significant confounders. While the environmental component in mouse recombinant inbred lines is controllable, their domesticated, unnatural genetic origin implies limitations in modeling outbred populations such as humans.

While there is an example of large-scale reverse genetics in *Drosophila* (PMID: 28084990), a wealth of targeted gene interference studies in zebrafish have highlighted many essential genes, mostly pertinent to monogenic phenotypes. There is one recent GWAS study on natural genetic variation and cardiac performance-related metrics using lines of the *Drosophila* Genetic Reference Panel (PMID: 36383075); however, with a significant phenotyping limitation for obvious reasons as it was based on manually dissected, disintegrated semi-intact *Drosophila* flies (PMID: 17360457). While zebrafish is the workhorse cardiac genetics fish model, providing straightforward phenotyping and a tremendous resource of mutant lines and

advanced functional gene interference methods, it lacks tolerance to inbreeding: it drops out for inbred approaches to study natural genetic variation. Instead, medaka provides this tolerance to inbreeding, and isogenic inbred medaka strains have been established from natural populations but have not yet been used for large-scale analysis of cardiac traits.

We agree that these considerations are central, not least to underscore the novelty of this study. As we indicated these points in detail in our recent publication on the Medaka Inbred Kiyosu-Karlsruhe (MIKK) panel (PMID: 35189950), we ensured that this reference, included in the original manuscript, is now placed more prominently in the introduction to provide a direct link for the interested reader.

As requested, we provide new *in vivo* experiments to underscore functional evidence and validate key findings.

We apologize for any confusion we may have caused and took advantage of the opportunity to clarify all of the raised points; answering them in detail below helped us improve the quality of the manuscript.

Specific comments

The introduction elides some of the core issues in quantitative trait definition and analysis. Indeed, cardiac physiologic and structural phenotypes, which are among the most common congenital abnormalities, are among the most useful exemplars of the dependence of genetic analyses on the resolution of phenotypic measurement. Traits which can appear continuous at the level of a population often become discontinuous as genetic mapping is undertaken (this has largely limited the ability to reduce QTLs to mechanistic variants in mouse or other species). Indeed, the examples where QTLs have led to important mechanistic insights have occurred where a specific QTL can readily be decomposed on more rigorous analysis into constituent binary traits, usually when formal segregation analyses are feasible at the relevant scale (PMID: 11907579). Thus, while ventricular size may appear as a QTL at a population level, it is clinically manifest as traits like HLHS a range of mitral, ventricular, outflow tract and aortic abnormalities with autosomal dominant transmission and reasonable penetrance but variable expressivity which is usually not detected except in formal proband-driven familial phenotyping (PMID: 29106500).

Indeed, this fundamentally superficial phenotyping is likely one contributor to the paucity of causal genes identified in congenital heart disease despite demonstrably high levels of familial recurrence for several of the most important syndromes (PMID: 29106500). Given the medaka trait they outline appears an interesting potentially tractable model for HLHS (including reduced survival to adulthood), it might have been more informative to have deeply phenotyped this strain and then mapped the genetic and environmental modifiers of survival in subsequent crosses.

We appreciate the reviewer highlighting the high relevance and challenges of quantitative trait genetics in nature and medicine. While we agree with principle considerations, as exemplified by the literature calls, HLHS specifically, supported by the current genomics data body, is not autosomal dominantly transmitted but has very heterogeneous genetics and is likely oligo/polygenic in nature, obviously with environmental sensitivity. However, the reviewer is pointing out a critical point here, i.e., that despite large-scale genomics efforts in human populations, there is a persisting contrast between elevated familial occurrence rates for several congenital heart disease entities, including HLHS and the relative inability to detect relevant genetic variants or genes as evidenced by the study of Ellesøe et al. 2018, called by the reviewer. As this paper is one among many convincing studies favoring a polygenic inheritance for most human congenital heart disease and also underlines the challenge of

phenotyping paradigms in patient populations imposed by the predominant discordance of cardiac malformations, we thank the reviewer for pointing out this essential reference and have included it into the introduction of the revised manuscript.

The introduction might usefully describe the limitations and confounding observed with quantitative traits and their subsequent mapping in other organisms. The complexity of true quantitative traits (as observed in the current models) is such that individual QTLs often break down in subsequent analyses where multiple genetic contributions within a single locus cannot be further reduced as the phenotype (quantitative trait variation) dissipates with any recombination within the locus haplotype. This has led to systems level analyses in other organisms for heart rate and other such traits. As noted above, when QTL approaches can be resolved through massive segregation studies for higher resolution phenotypes, there is the potential to resolve QTLs at a mechanistic level.

In addition to our remarks to “General comments” (see above): We perfectly agree that large-scale segregation crosses are a particular strength of model system mapping approaches, which is impossible in humans. We are aware of e.g., the *Drosophila* panel and work based on classical inbred mouse strains, with inherent limitations due to historical breeding regimes in terms of wild genetic variance coverage, and introduce with our study in medaka a scalable vertebrate inbred genetics approach, demonstrated using a subset of currently available strains with a view on full expandability provided by our recently published medaka inbred resource *The Medaka Inbred Kiyosu-Karlsruhe (MIKK) panel* (PMID: 35189950 and PMID: 35189951), as quoted in the manuscript. It allows employing the potential of wild polymorphisms for QTL dissection of cardiac traits, which has not been done so far. Using medaka inbred strains as a mapping resource, we demonstrate the high degree of phenotyping reliability, cost-effective crossing, sequence and mapping strategies, and direct routes for functional validation using state-of-the-art genome editing techniques, which we are convinced is a unique and valid approach for cardiac genetics.

The section entitled heart phenotype contrasts also seems to elide different attributes of heart rate and cardiac function which are already extremely well understood. For example, there is no discussion of the effects of extracardiac physiology (e.g. neuronal, skeletal muscular or metabolic) on heart rate, even during early development. Cardiac structural morphology is a broad descriptor for a wide range of abnormalities with discrete effects on heart rate and cardiac function presumably refers to contractility, relaxation etc, which also have some competing influences on heart rate through very discrete mechanisms and which might have been discerned in parallel during assessment of the parental strains and would have assisted in choosing the optimal experimental design to explore heart rate as a ‘pure’ quantitative trait.

In the same vein of the interrelatedness of different physiological layers, as correctly pointed out by the reviewer, we used heart rate metrics intentionally as it is reflecting interactions between different cardiac properties “including electrophysiology, morphology, and function.” Moreover, as published in our previous method paper (PMID: 32029752), we established an automated imaging pipeline, allowing for reliable and precise phenotyping, accomplishing one precondition to tackle quantitative genetics. Of note, profiling embryonic heart rates in different medaka inbred strains led us to identify not only differences in heart rate levels but also significant quantitative morphological differences, i.e., hypoplastic ventricle in the HO5 strain, demonstrating the robustness of the heart rate parameter reflecting multiple phenotypic layers with developmental manifestation. Consequently, mapping heart rate metrics implies

association with genes that may primarily affect heart rate but also impact cardiac development and function through different mechanisms, which we accounted for in the validation layout and the result and relevance of which we tested by direct state-of-the-art gene editing techniques.

It is not clear that plate address was controlled for in the data acquisition and analysis but even in a regulated chamber the light necessary for video acquisition has led to temperature differences which over a 96 wellplate can be substantial.

The referee is right, this is an essential point. Apparently we failed to sufficiently highlight that this has been critically assessed in a previous manuscript describing the assay (PMID: 32029752). In short, the microscope (Acquifer imaging machine) uses LED illumination and a temperature-controlled imaging chamber and we have carefully controlled for well to well differences in temperature and heart rate in the preparatory research.

We have taken care to highlight and prominently reference this point in the revised version of the manuscript, adding respective comment in the Figure1 caption in addition to our method description of the initial manuscript:

“To exclude positional effects of plate coordinates on heart rate, a full 96-well plate with Cab embryos was recorded at 28°C. Heart rates were normally distributed ($P=0.86$; Shapiro-Wilk normality test). One-way ANOVA indicated no significant effects of plate row on mean heart rates (degrees of freedom (df) = 7.88; $P = 0.61$), nor of plate column on mean heart rates ($df = 11.84$; $P = 0.15$).”

The authors note discrete swimming performance between strains but fail to discriminate between cardiac and skeletal muscle contributions to this phenotype. Is the heart rate phenotype in the HO5 strain purely cardiac or is there also a skeletal muscle abnormality. The failure to account for orthogonal phenotypes renders subsequent heart rate analyses difficult to interpret.

We thank the reviewer for raising this point. The heart rate assay was performed in embryos, which were still naturally constrained in the native chorion. Thus, swimming behavior does not apply here. However, we agree with the reviewer that a potential skeletal muscle phenotype could impact adult swimming behavior. Following this valuable suggestion, we generated and analyzed skeletal muscle vibratome sections. Microscopic analysis of skeletal muscle sections revealed histologically intact skeletomuscular organization in adult HO5 fish, thus not providing evidence for a primary skeletal muscle phenotype underlying the poor swimming performance. However, the constitution of adult HO5 (relatively short and round fish) might also play into the movement pattern and performance. At the same time, our echocardiography readout demonstrates low cardiac output, which is likely directly related to physical fitness. We modified this section in the revised manuscript to put these observations into the context of additional histological analysis, provided in new Figure S1.

Segregation analysis

The authors perform an F1 intercross and then sample ‘1260 F2 individuals unique inherited genotypes’. It is not clear what this sentence means. Were genotypes used to dictate the sampling? Were haploid fish generated and then offspring sampled or is this just 1260 F2 individuals all of whom as in any F2 would be assumed to have “unique inherited genotypes” on the basis of normal meiosis?

Important questions - thank you. Genotypes were not used to dictate sampling. We collected 1190 F2 in an unbiased manner which were subjected to genotype/phenotype correlation. We changed this sentence to "(...) F2 generation with unique recombined genotypes" in addition to the Caption to Fig. 1A in the initial manuscript: "Crossing setup used to generate HdrR × HO5 offspring with segregated SNPs in the second generation: isogenic HO5 and HdrR parents are crossed to generate hybrid (heterozygous) F1 generation (grey) with intermediate phenotype, which after incrossing results in F2 individuals with individually segregated SNPs resulting from one cycle of meiotic recombination."

Genome-wide QTL mapping

The authors then undertake a standard QTL analysis using the recombinant panel they have created among the 1260 F2 offspring. The rationale for now exploring temperature sensitivity is unclear when these data are not presented for the parental strains.

Apparently, we failed to clearly introduce the mapping strategy. In the revised version of the manuscript we have made an effort to clarify this critical prerequisite. Embryos of the parental inbred strains were used to assay for strain-specific differences in heart rate at three given temperatures (numbers per strain, variance, etc). As a result we identified the HO5 and HdrR inbred strains exhibiting significant differences in heart rate when comparing the strains at 21, 28 and 35C. We profiled the temperature sensitivity of heart rates of the two parental strains, HO5 and HdrR and then performed the heart rate screen also across the three test temperatures in the F2 population (Fig. 2). We have clarified this in the revised manuscript.

There is a clear over representation of HO5 alleles which is somewhat concerning given the discrete gross phenotypes and the discrete viability of this strain as described. This is attributed to the training set used for the Markov Model, but the potential plausible biological causes could also have been formally tested with a single additional cross.

Although the imbalance in alleles could potentially be a result of incompatibility between certain genomic regions between HO5 and HdrR, this is not the focus of the study.

The issues with QTL mapping are highlighted nicely in the analyses they outline. Without discrete recombinant mapping to much higher resolution, one has difficulty in identifying the transmissibility of the trait components associated with individual haplotype blocks and the variants or genes involved cannot predictably be defined. Indeed, all of the issues with association mapping are amplified here because both trait imprecision and locus imprecision compound. Trait segregation patterns cannot be rigorously defined and no part of the locus can be discretely associated with the trait (it could readily be an intergenic regulator of genes far outside of the block associated with the trait). To overcome these challenges one could continue to scale the generation and analysis of informative meiotic events, but as is typical in such studies the jump is made to use less rigorous (and less costly and time-demanding) approaches including gene expression analyses or testing of generic LoF alleles both of which make quite substantial assumptions about mechanism which are never then tested. Examples from yeast are particularly informative in this context where single 'QTLs' may be resolved into discrete gain of function alleles only a few base pairs apart with very large effect sizes that are simply undetectable without rigorous segregation to variant level (PMID: 11907579). The potential to execute on this in a vertebrate is at present largely restricted to fish species and might have been usefully undertaken with one additional cross.

The gene enrichment analyses are perfectly well executed and described. Their relevance to the remainder of the manuscript is largely by word association rather than experimental proof. The authors assume that the genes responsible for their heart rate observations are differentially expressed in a particular time window in a single tissue and are physically located within recombination boundaries that are statistically defined.

We thank the reviewer for mentioning this section. We appreciate certain limitations of transcriptomic data and have therefore included multiple criteria to pinpoint candidate genes of potential relevance. Indeed, we demonstrated successful short listing by gene enrichment providing an excellent entry point for future mechanistic downstream studies. The data are flanking the genetic mapping and are used as supportive evidence. The experimental validation of these differentially expressed genes is beyond the scope of the work presented here.

In vivo validation

The authors test CRISPR LoF in a series of candidate genes defined by the locus boundaries and the gene enrichment analyses. There is no formal demonstration of the mode of transmission or of any other analyses which might support such a model. How were gain of function effects excluded?

While CRISPR-mediated editing allows controlled gene targeting, introduced mutations are not per se classified as loss-of or gain-of function. We have targeted the interference with the identified candidates to maximize the impact (e. g. conserved, crucial functional domains, introduction of premature stop codons) to be able to assess the putative contribution of the large number of candidates presented. The detailed mechanistic analysis will be the focus of future work. We have made sure to eliminate misleading terminology in the revised version of the manuscript and have avoided the term “loss-of-function mutations”.

Many of these issues (which affect all such QTL studies) could have been at least partially addressed by closing the phenotypic loop for the heart rate traits. For example if there are heart rate phenotypes with a specific gene LoF which are mediated via discrete effects on variable AV delay this could then be studied in the relevant parental strains to see if discrete mechanisms are transmitted at a phenotypic level even if not formally shown at a genetic level. Allusions are made to different mechanisms for human traits associated with some of the genes but none of these mechanisms (automaticity differences, conduction block, repolarization prolongation, etc) are then tested in the medaka model.

Using natural variation present in the F2 mapping population allowed to link heart rate metrics to QTLs and ultimately new candidate genes. While those markers establish the mapping, our intention of the CRISPR-based validation was to uncover foremost the full phenotypic potential of a given test gene. For example, we linked *rrad* and then uncovered a specific conduction phenotype in gene editing experiments, secured by clear genotype-phenotype correlations stable over generations. Irrespective of deep phenotyping, reserved for future work, this *in vivo* finding is particularly relevant given the association of the human ortholog *RRAD* as a new susceptibility gene detected in familial Brugada syndrome cases (Belbachir et al. 2019, PMID: 31114854), with cardiomyocytes derived from induced pluripotent stem cells expressing pathogenic *RRAD* variant exhibiting electrophysiological features of Brugada syndrome and cytoskeleton disturbances. The arrhythmic manifestation of Brugada-associated gene mutations in a two-chambered fish heart is obviously different in terms of anatomical location of arrhythmogenic areas as compared to a four-chambered human heart. As such, the exact recapitulation of the electrophysiological phenotype is not the primary

motivation for studying gene function in the medaka model but rather for testing the general mechanistic role and *in vivo* relevance of a gene of interest.

Moreover, while Brugada syndrome is known for its propensity to ventricular arrhythmia, slow electrical conduction in subareas of the heart is thought to represent a major electrical pathological substrate. In this regard, it is a particularly relevant finding that *rrad* knockout models in medaka showed atrioventricular blocks, likely as a manifestation of a slow conduction phenotype. Additional significance to this finding is added by observations that Brugada patients can develop severe (high-risk) atrioventricular blocks (AVB); intrinsic conduction abnormalities were discussed as one of the potential mechanisms for AVB, while the genetics in Brugada patients at risk remain elusive (Kamakura et al. 2021, PMID: 33428312). We provide new *in vivo* evidence for *rrad* mutations leading to AV block, which could have immediate prognostic and therapeutic value (irrespective of the actual subcellular mechanism) for a subset of Brugada patients, as Brugada patients with high-risk AVB should be managed by cardiac pacing (which has implications for the choice among different types of implantable cardioverter defibrillators). We took advantage of this reviewer's input and restructured and refocused the related section in the discussion.

The authors might consider how likely the average gene not selected by the process they describe would be to have a generic effect on heart rate. Such a control group could have been tested for each step (genetics, gene enrichment and *in vivo* testing) for generic heart rate or for high resolution mechanistic phenotypes. The current manuscript does not include any such controls.

We agree it is essential to test a hit rate at baseline, which was completed by independent work from our lab, which we had referenced (please cf. reference 48 of the main manuscript) in the initially submitted version. We made sure in the revised manuscript that this important control is more prominently put to the reader's attention.

Discussion

The discussion is perfectly reasonable but glosses over the methodologic challenges noted above. These challenges are undoubtedly not unique to this study but the core message of the manuscript is that somehow access to fish radically changes the rigor of the genetic dissection of human disease traits by "partitioning and validating the spectrum of phenotypes in a strain specific way". The manuscript highlights the confusion that appears to exist in modern genetics between genotype and phenotypic architecture. As noted in the comments on the *in vivo* validation section, one thing that this manuscript does not even attempt is to shed any light on the partitioning of phenotypes. Among the missed opportunities in the current study is the chance to resolve discrete strain-specific effects on a high-level trait like heart rate into discrete phenotypes and then to map these to individual variants by which mechanism might be discerned.

We fully agree with the referee that there is a plethora of follow up experiments to tackle the details of the underlying mechanisms. What we provide is the framework for this future analysis, which is not necessarily limited to medaka in particular or fish more generally. With our work we uncover and validate evolutionarily conserved key players that on together contribute to a complex phenotype. However, depending on the genetic context they can exhibit more or less severe effects. This is best reflected by the strain specific genetic variants identified and discussed in the revised manuscript. We discovered a LoF variation as a single premature stop codon variant in the *blzf1* gene in the HO5 strain with relatively mild consequences. When a premature stop codon is introduced into the orthologous gene in the

HdrR background it displayed a very pronounced impact on heart morphology and heart rate highlighting the impact on heart morphology and heart rate and underpinning the complex genetic interactions selected for in the specific strains. With the experimental pipeline presented and the toolbox developed our work indeed opens the door for addressing the mechanistic basis in a follow up study.

Point by point response to the reviewers' concerns:

Reviewer #1 (Remarks to the Author):

The authors have addressed my comments with regards to the written part of the manuscript.

They have also addressed most of my concerns and questions regarding the missing experimental data.

However, I am a bit uncertain as to why the mutant line carrying a mutation in RRAD, the most promising gene from a human disease point of view, has not been characterized farther?

Since this gene has previously been associated with severe myofibril disarray (PMID: 33195237) in cultured cardiomyocytes, as well as calcium mishandling, it is surprising that the authors have not included the mutants in the sarcomere analyses done on the other 4 mutants.

The myofibril disarray would be a simple add-on to show that the in vivo knock down recapitulate the phenotype observed in cell culture, and add to the significance of the screening approach the authors present in the current study.

We thank the reviewer for his final positive evaluation. We appreciate his note on RRAD, which further stresses the relevance of our findings. We point in the manuscript to another study (PMID: 31114854), which also shows cytoskeletal defects in cardiomyocytes derived from Brugada syndrome patients, in addition to the reviewer's reference. While these structural defects were expected, we focused on the novel finding of arrhythmia (AV block) in the medaka rrad knockout model.

Referee II:

The authors have responded to each of the reviewers' comments, and while they have undertaken some additional experiments, the bulk of their responses are simply restatements of their original output with some additional results but no new supportive data. The data presented simply do not support the conclusions they have drawn. Specific comments are outlined below.

We thank the reviewer for his efforts to evaluate our revision.

Unlike his general assessment that our revision is merely restatements of our original output, our response to his questions and concerns has been thorough, addressing theoretical issues point-by-point and incorporating additional experimental data. As per one of the key requests of both reviewers, we analyzed the skeletal muscle structure of the parental lines to discriminate the cardiac phenotype from a potential skeletal muscle phenotype to substantiate further the cardiac phenotype contrast in the parental lines that served as the basis for the F2 linkage mapping. Additionally, we provide clear evidence for a functional role in myosin formation for a selected group of candidate genes initially identified as regulators of cardiac ventricle size through the morphological assessment of crispants. This was achieved by conducting new CRISPR-mediated gene editing and high-resolution imaging, confirming the direct involvement of these genes in the growth of the ventricular chamber.

Thus, the reviewer's statement that "the bulk of their responses are simply restatements of their original output" lacks justification, as elaborated below.

The lack of novelty is addressed by reframing their initial claims so narrowly so as to reinforce the original comment. In essence they note that this is the first study with two naturally originating isogenic Medaka lines (as for all prior mouse and fly studies).

This needs to be corrected. We noted that this study utilized five naturally originating isogenic medaka lines, focusing on two due to their significant phenotypic differences, constituting valid phenotypic and genomic properties for an effective F2 mapping approach. We never claimed this is "the first study [...]". However, it is indeed the first study using isogenic lines that investigates quantitative cardiac traits at a genome-wide level in terms of high-throughput imaging and sequencing and a large F2 population size. This aspect is not merely stated in the manuscript; instead, it emphasizes the strategy and the data, highlighting the study's novelty.

Efforts to detect and explain the genetic determination of heart disease, particularly congenital heart disease (the most common congenital/developmental type of defect), still face shortcomings in human studies. We provide an alternative strategy using the medaka model not previously employed for this purpose and at this scale to detect and validate new genes with human relevance, thus synergizing with data derived from human populations.

They also somehow suggest that 'naturally originating' isogenic lines are more representative of outbred human data. Obviously, the prior studies with isogenic mouse and fly genetic studies have as much relevance to observations in outbred humans as the current dataset since they also are originally derived from naturally occurring alleles.

Throughout the manuscript, we have deliberately refrained from comparing the medaka model with mouse or fly, or positioning it as more representative of outbred human data. In fact, to reinforce the strength of inbred genetics as complementary approach to study human-relevant phenotypes, we do cite Trudy Mackay's landmark inbred *Drosophila* paper (Mackay, T., 2012) when we state that "The F2 cross of inbred strains with contrasting phenotypes allows to segregate causal alleles in a way that cannot be achieved with human GWAS studies" (Discussion).

Our manuscript is not about comparing different model systems, but rather about highlighting the new insights enabled through advantages of the medaka model, including inbreeding efficiency, amenability to phenotyping, sequencing pipelines (individual phenotyped specimen in a 96 well format) and direct CRISPR manipulation.

We believe that all model systems have their own strengths, and that the combination of different model systems or human-derived data is the foundation of continued progress in the field. Here, we are adding another significant study to this body of knowledge, focusing on an inbred vertebrate model with exceptionally high scalability compared to mouse (we do not make any superior statement in the manuscript). Our study will interest a broad readership, not least for its applicability to any quantitative phenotype for which reliable assays can be developed.

However, we value the reviewer's acknowledgement that "the current dataset" has "as much relevance to observations in outbred humans" as "studies with isogenic mouse and fly genetic studies".

While our paper does not intend to review different model systems, we note at this point that the origin of the laboratory mouse strains is due to the forced breeding of a number of separated species in Japan and Boston in the 19th and 20th centuries, meaning the presence of segregating alleles in laboratory animals is not of the same origin as segregating alleles from a natural population.

They also confound quantitative traits in human GWAS (where there are loci associated with contributions to such traits through linkage disequilibrium but no evidence of transmission) with QTLs in an incross (where the transmission and segregation enables locus definition using recombinants).

There is no such confusion. Throughout the manuscript, we outline QTL linkage mapping ("linkage blocks", "linkage probabilities", "strength of linkage") that enabled us to identify candidate genes. Based on orthology, these candidates were cross-correlated to human studies, mostly GWAS, enhancing human relevance.

Finally, the authors suggest they have produced new in vivo experiments, but it is not clear to this reviewer to which experiments they refer. There is a pervasive failure in the manuscript to distinguish between genomics and genetics. For example, it is simply not possible to define the mode of inheritance from single generation genomics.

We do not define "the mode of inheritance" - we used a F2 segregation cross for QTL mapping of genes with potential relevance for cardiac biology, and ultimately demonstrated functional relevance for select candidate genes utilizing CRISPR-based gene editing.

The PMID citation noted in my original comment clearly demonstrates the inheritance mode of HLHS (among other congenital heart disease traits) is autosomal dominant from familial

aggregation studies. Indeed, I cited this study precisely because it highlights how lack of rigor in phenotyping (ie assuming that every carrier of the causal genotype in the kindred will have exactly the same phenotype as the proband) underlies the inability to map and close these conditions in humans (as in medaka and any other species).

We want to remind the referee that we had extensively replied to this point in the first revision. Additionally, we note here:

The crux of our study is not determining the mode of inheritance for hypoplastic left heart syndrome (HLHS) but emphasizing that while there is a clear indication for a genetic etiology based on familial clustering (PMID:30571578), epidemiological and recent experimental data indicate a complex genetic model (PMID:34694888). The misconception of the reviewer that HLHS is inherited in an autosomal dominant manner, perpetuated by a misguided reliance on a single epidemiological study on congenital heart disease (PMID:29106500) that only marginally touches on HLHS as one of many entities and ultimately fails to support the reviewer's claims, as there is predominant discordance for HLHS family rate (Figure2 in PMID:29106500). Such oversights are not only misleading but risk distorting the foundational understanding of HLHS.

It is crucial to clarify that HLHS is a complex, multigenic condition (PMID: 28530678), as mischaracterizing HLHS as an autosomal dominant disorder would undermine the entire manuscript and the insights it provides. Our approach utilizes the inbred genetics of medaka, harnessing its natural genetic variation to probe the polygenic influences at play. While the significance of this polygenic contribution cannot be overstated, it remains challenging to dissect it in human populations (incomplete penetrance, variable expressivity, GxG, GxE, along with individual genetic modifiers). In this context, employing inbreeding-tolerant medaka strains emerges as a powerful model, offering an extraordinary opportunity to conduct precise phenotyping within vertebrate isogenic genomic backgrounds.

Moreover, our ability to implement advanced gene editing techniques within the same model system - demonstrated by established methodologies from our lab (e.g, PMID:35333175, PMID:35373735, PMID:37584388) - provides a direct functional readout of genetic contributions. This synergy between mapping conserved genes and validating their functions in medaka significantly enhances the existing data derived from human studies. We have highlighted instances where genes previously loosely associated with human genome-wide association studies (GWAS), were now rigorously validated in our study by recapturing heart-related genes in a different genomic context and through functional validation, thereby underscoring the relevance and impact of our findings.

Their interpretation of these findings highlights the lack of discrimination between familial aggregation and transmission likelihood on one hand and measured variation across the genome on the other. With the degree of familiarity exhibited the mode of inheritance cannot be polygenic under normal transmission probabilities.

It is unclear what is being referred to by "the degree of familiarity exhibited" however, we account for relatedness structure in our GWAS tests using a linear mixed model and given the inheritance patterns observed in the F2 population from isogenic F0 founder lines it is feasible that our mapping procedure can uncover GxG effects and polygenic traits.

Although the reviewer is right that a F2 cross has a more limited number of causal segregating loci than an outbred population, it is entirely expected that for polygenic traits they will remain

polygenic in F2 or advanced intercross settings. For example, in this mouse intercross ([10.1534/genetics.114.167056](https://doi.org/10.1534/genetics.114.167056)) the majority of traits are polygenic with >1 locus, and indeed the fact that a number of loci were above our permutation threshold in this study shows we were able to discover >1 locus. In addition when we performed a linear mixed model with no fixed loci, we estimated narrow sense heritability as 0.465% whereas the sum of the loci discovered provided 0.123%

I agree that the potential for Medaka to resolve true QTLs (if they can be resolved at all) is relevant, my point is simply that the current manuscript does not resolve these to any greater extent than prior mouse or fly data despite the potential for definitive resolution through exactly the deep phenotyping which I suggested in my review.

To be more precise, one hypothesis for the difficulties in resolving QTLs is the focus on a single quantity. In other words if one were to explore heart rate in conjunction with other orthogonal traits such as ventricular mass, AV conduction or Ca²⁺ handling, then one might be able to define robust genetic support (as opposed to genomic association) for specific variants in specific genes and then model how these various loci interact to generate a final output (see also below on Brugada claims).

Our study aimed to identify “genes influencing quantitative cardiac phenotypes” by leveraging heart rate as a robust, high-throughput readout that can be determined as a non-invasive measure in fish embryos. Heart rate phenotypes, as the reviewer noted, can originate from diverse mechanisms such as “ion channels, calcium-handling proteins, and junctional molecules”. This diversity makes heart rate an ideal entry point for uncovering multiple genes with broad relevance to the cardiovascular system, without unnecessarily constraining the discoveries to a single mode of action.

The results of this study clearly demonstrate the power of our approach to uncover novel genes with critical roles in cardiovascular function. Building on our findings, future studies will expand the phenotypic assessments to additional parameters - such as “ventricular mass, atrioventricular conduction, or calcium handling” - while integrating genome editing for comprehensive validation. This will further extend the impact of our findings and deepen insights into cardiovascular gene function.

While the suggested follow up and in-depth phenotyping is fundamentally sound, any analysis crucially relies on resources that can only be invested once the proof of concept study (our work presented here) has demonstrated the fundamental applicability of the chosen approach. Respecting the guidelines of our funders we have decided to stick to that order.

Genome editing is just that. Individual alleles may have gain or loss of function or both for the gene of interest. Knocking out a gene with CRISPR is helpful validation if you have data that the trait which you are trying to model has the relevant inheritance characteristics which suggest dose dependent loss of function, but many individual mutations even in large effect size Mendelian disorders have quite clear effects through loss of some functions and gain of others. Otherwise, gene editing simply shows that the gene is involved in some way in cardiac biology-in particular if the phenotype is so superficial as simply any effect on heart rate. Even the sodium channel gene which the authors refer to in their discussion of Brugada syndrome exhibits variation which has effects on multiple different functions (conductance, coupling, excitability, force transduction) in different disease syndromes.

We stated that “we focussed on prominent candidates that we specifically edited in their genomic context to address and validate their potential role in the development of cardiac phenotypes”. While editing candidate genes uncovered a distinct impact on cardiac phenotypes, this was expected as different layers of phenotypes related to the heart rate used at the F2 phenotype. The primary role of our CRISPR experiments was to robustly demonstrate the in vivo candidate gene’s significance for “cardiac biology” but not to disentangle any possible mechanistic gain or loss of function mode, which is only to be considered in focused downstream studies to model different mutations when zooming in on specific human variants with predicted loss of or gain of function effects.

The response which invokes physiologic layers further highlights the problems outlined above. The authors assume that there is a discrete relationship between heart rate and the other layers they cite, but my point is that there are multiple different ways to generate any particular effect on heart rate and without looking at the other layers, one is forced to assume that each heart rate has some distinct validity which is already known not to be the case (see multiple ion channels, calcium handling protein and junctional molecules which result in slowing of heart rate in human disease yet can all be discriminated by associated orthogonal phenotypes). The authors appear to recognize these limitations but suggest that they are overcome by “direct state of the art gene editing techniques” when they can only be resolved by extensive additional phenotyping.

As mentioned above, our study aimed to identify candidate genes that contribute to cardiac phenotypes and provide an initial characterization as a proof-of-principle approach that validates the discovery process, thus underlining the relevance and potential of our findings. The choice of heart rate as a core phenotype for the actual mapping has been extensively elaborated in the previous revision.

The appropriate statistical approach for assessing ‘within plate’ variance in screening is a formal z-score.

We demonstrated that there are no positional effects on plate coordinates on heart rate (cf. revision1; see Gierten, et al. 2020, 10.1038/s41598-020-58563-w). Still, we use plate variables as a covariate in our genetic association and other studies, similar to many other F2 and genome wide association studies. This allows this covariate to be fitted in the regression with each locus of interest.

Among the new data in the revised manuscript appear to be skeletal muscle histology in the H05 adults. Obviously, this does not address functional performance in any direct way, but also highlights the generally superficial nature of the phenotyping rationale across the manuscript. Low cardiac output is also observed but the presence of this one phenotype does not preclude any or many others.

In the first revision skeletal muscle analysis was requested. Now our response to the referees request is taken as indication for superficiality. With our histological analysis we addressed the reviewer's concern regarding "skeletal muscle contributions to the discrete swimming performance" between the HO5 and HdrR strain that could be based on "skeletal muscle abnormality". The regular muscle organization in HO5 has not resulted in any evidence supporting or justifying additional animal experiments to address functional performance.

The authors do clarify their intercross and genotyping strategy while also noting prior data on temperature sensitivity in the parental strains.

We thank the reviewer for the input that helped to clarify the conducted study workflow in the text of the manuscript.

The authors agree there is overrepresentation of the H05 allele but somehow feel this is not the focus of a study whose fundamental conclusions are based on allele transmission with heart rate. This might be a central finding of the study-if there is some gene-gene interaction which favors H05 (possibly a survivorship bias or selection of HLHS prone alleles due to some developmental constraint) it might be the key to the presence of such potentially lethal alleles in the human population). The lack of follow up of this genotype imbalance substantially weakens any conclusions.

We agree with the reviewer that it is interesting that a number of loci show biased allelic transmission, but there are a variety of hypotheses that could explain this, from global hybrid incompatibility - not uncommon in inbred line F2 experiments across a number of organisms - through to specific meiotic drive. Follow up of the genotype imbalance is interesting but we rebut the idea that the genotype imbalance weakens the conclusions, in particular in light of the genome editing follow up of loci of interest, something that many robust findings of F2 crosses in plants and animals do not have not least because of many of them predate genome editing technology. For the loci we discovered they were not strongly biased suggesting that the overall allelic biases we observed did not influence the traits of interesting.

The authors note the challenges with their transcriptomic data but have included these data without validation.

Importantly, we did not note challenges relating to the transcriptomic data. As outlined in the manuscript, the results from the gene expression analysis were used to prioritize and select candidate genes for further experimental validation to complement the QTL mapping data.

Surely the generic modeling of arbitrary alleles rather than the specific mechanistic effect of the naturally occurring alleles confounds the premise by which the authors claim novelty for their model and specific experiments. It should be noted that the generation of isogenic lines already selects for very specific subsets of mechanism.

This comment equally applies to any genetic model, fish as much as flies, and mice. Naturally occurring alleles fixed in isogenicity have the potential to drive phenotypes towards the extremes of physiological distributions, which we demonstrate at the levels of high heart rate and small chamber size (HO5). Genes carrying such variants have been ultimately detected through post-segregation mapping and experimentally validated. The more lines

are used for the initial screening (scalability for future studies outlined in the manuscript), the more variants, i.e., “mechanistic effects”, can be subjected to the gene discovery process. Functionally modeling specific mechanisms of distinct alleles is part of detailed downstream studies of genes that have been put forward in this study relevant to different cardiac developmental and functional processes.

It follows that looking at fish genetic variants in fly, fish, and mouse species will definitely widen and enrich the spectrum of contributions.

The section on RRAD further highlights the laxity in phenotyping and terminology which characterizes the current manuscript. The authors note that regional conduction abnormalities are seen in the ventricles of Brugada subjects but then suggest that AV conduction (a completely distinctive developmental tissue not shown to be involved in Brugada related AV block which is where reported infraHisian and probably junction related rather than channel conductance related, though there are other sodium channel disorders which are associated with AV block-once again all confounded by low resolution phenotyping in both human and in this instance Medaka) represents a surrogate for this highly specific disease phenotype. This could be true, but AV block per se does not contribute to ventricular arrhythmogenesis in Brugada syndrome, and the authors also assume that the AV block observed is orthologous without any new data.

We want to clarify the considerations we stated in the first revision to capture the relevant points beyond textbook knowledge. We detected RRAD association with heart rate and noted that this is a new susceptibility gene detected in familial Brugada syndrome cases (Belbachir et al. 2019, PMID: 31114854) with cardiomyocytes derived from induced pluripotent stem cells expressing pathogenic *RRAD* variants exhibiting electrophysiological features of Brugada syndrome and cytoskeleton disturbances. We also noted obvious fundamental differences between the two-chambered fish and four-chambered human heart, yet there are many high-quality papers demonstrating the power to study genetics and pathophysiological principles of human channelopathies in fish models. In this regard, we acknowledged that while the classical pathological hallmark in humans is a ventricular conduction delay, it was beyond the scope of this study to test ventricular conduction in our *rrad* knockout model specifically.

However, in this knockout model (germline transmitted), we observed a consistent AV block, which we did not directly compare to slow ventricular conduction in humans. Still, we referred to the published observation that Brugada patients can develop severe (high-risk) atrioventricular (AV) blocks. Brugada syndrome patients are known to exhibit variable conduction abnormalities, including the QRS complex (ventricle) and PR interval (AV conduction) (PMID: 25016148). In the case of the human AV block phenotype, which is an important prognostic marker (PMID: 29986018), intrinsic conduction abnormalities were discussed as one of the potential mechanisms (Kamakura et al. 2021, PMID: 33428312), while the exact anatomic (bundle or higher) and molecular substrates remain unresolved.

The genetics of Brugada patients at risk of developing severe AV block remain elusive. In light of the reportedly complex genetics, we stated that our *in vivo* evidence for *rrad* mutations leading to AV block is a particularly medically relevant finding.

Collectively, we did not compare an AV block in fish with ventricular arrhythmia in humans but drew a potential relationship between human AV block in Brugada patients (PMID: 33428312) and the AV block displayed in the *rrad* knockout fish model.

There is also an effort to characterize the effects of editing 3 genes on cardiomyocyte structure as highlighted by some of the data presented for *adprhl1*. It is not obvious how this assists the message of the manuscript since this is but one such potential phenotype among many which might have been tested.

Due to the proposed role of *adprhl1* in myofibril assembly and the reviewer's request for deeper phenotypic investigations underlying the observed heart rate and heart morphology phenotypes we acquired high-resolution imaging data of the sarcomeres. The reduced myosin light chain signal and sarcomeric disarray in *btbd1* and *adprhl1* crispants reveals further insights into the potential role of these genes in cardiac function and by that highlights that the heart rate is an effective and reliable readout to identify genes underlying cardiac phenotypes.

The authors note that the control data were not collected in parallel but rather date from a prior set of experiments. This essentially renders much of the work presented difficult to interpret given the effects of selection and the wide range of mechanisms by which heart rate might be affected.

The referee's assumption is not correct. As stated in the manuscript, gene editing effects of all conducted validation experiments were always compared to mock injected control embryos that were acquired in parallel within the same experimental series. In the initial submission we included only the heart rate data of "embryos with apparently normal heart development" (Fig. 4). Upon request by reviewer 1 we provided in addition the heart rate data for "embryos with aberrant cardiac morphogenesis" acquired within the same single experiment (Fig. S6). Thus, the mock injected embryos not only derived from the same egg collection batch, but also were injected and phenotyped on the same days as their edited siblings.

The final allusion in the authors' response to reviews is that they are simply setting the stage for subsequent work. As noted, the results of these downstream experiments would directly address the uncertainty currently clouding their conclusions which remain incompletely substantiated by the data presented.

As mentioned above, the aim of our study was to identify candidate genes that contribute to cardiac phenotypes and to provide an initial characterization as a proof-of-concept approach that validates the discovery process, thus underlining the relevance and potential of our findings. While the suggested following up in-depth phenotyping is fundamentally sound, its substantial requirements in terms of time, cost, and labor prompted us to not include it as an additional scope of our manuscript.

Reviewer #2 (Remarks on code availability):

There are literally tens of such algorithms in the literature (even just for fish)-the majority based on the same basic mathematical transform.

The reviewer is absolutely correct. Here we respond to the request of the Journal to stick to transparency, reproducibility and availability of the code. The point reflects a response to the Journals requirements and is not meant to indicate novelty.